# AP-LDM: Attentive and Progressive Latent Diffusion Model for Training-Free High-Resolution Image Generation

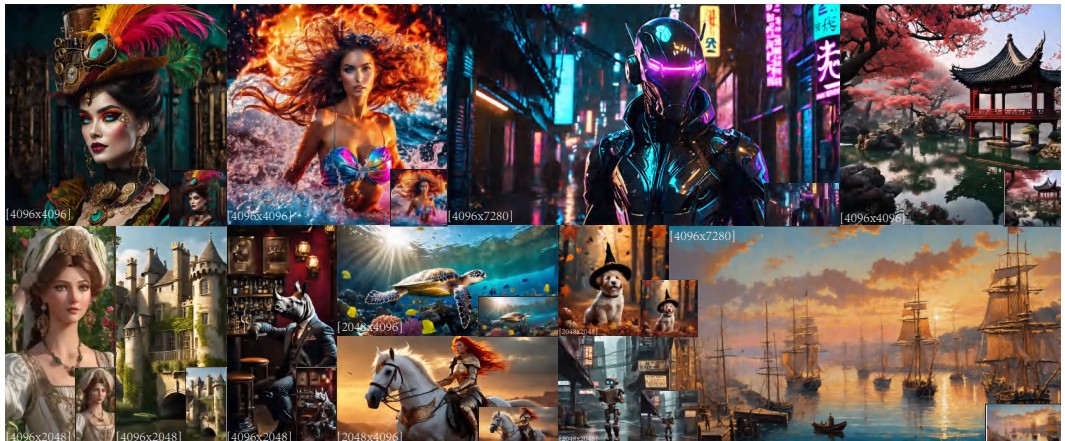

Figure 1: **High-resolution images generated by our AP-LDM using a single 3090 GPU.** The corresponding thumbnails are generated by SDXL (Podell et al., 2023) at their training resolution.

## Abstract

Latent diffusion models (LDMs), such as Stable Diffusion, often experience significant structural distortions when directly generating high-resolution (HR) images that exceed their original training resolutions. A straightforward and cost-effective solution is to adapt pre-trained LDMs for HR image generation; however, existing methods often suffer from poor image quality and long inference time. In this paper, we propose an Attentive and Progressive LDM (AP-LDM), a novel, training-free framework aimed at enhancing HR image quality while accelerating the generation process. AP-LDM decomposes the denoising process of LDMs into two stages: (**i**) attentive training-resolution denoising, and (**ii**) progressive high-resolution denoising. The first stage generates a latent representation of a higher-quality training-resolution image through the proposed attentive guidance, which utilizes a novel parameter-free self-attention mechanism to enhance the structural consistency. The second stage progressively performs upsampling in pixel space, alleviating the severe artifacts caused by latent space upsampling. Leveraging the effective initialization from the first stage enables denoising at higher resolutions with significantly fewer steps, enhancing overall efficiency. Extensive experimental results demonstrate that AP-LDM significantly outperforms state-of-the-art methods, delivering up to a $5\times$ speedup in HR image generation, thereby highlighting its substantial advantages for real-world applications.

## 1 Introduction

Diffusion models (DMs) have demonstrated impressive performance in visual generation tasks, particularly in text-to-image generation (Ho et al., 2020; Nichol & Dhariwal, 2021; Podell et al., 2023; Esser et al., 2024; Mou et al., 2024; Zhang et al., 2023; Feng et al., 2023). One notable variant

Figure 2: **Comparison of our AP-LDM with prior work in generating** $2048 \times 2048$ **image**. The prompt is *Neon lights illuminate the bustling cityscape at night, casting colorful reflections on the wet streets*. Zoom-in for a better view.

of DMs is the latent diffusion model (LDM), which performs diffusion modeling in latent space to reduce training and inference costs, enabling high-resolution (HR) generation up to $1024 \times 1024$. While it is possible to modify the input size for higher-resolution generation, this often results in severe structural distortions, as illustrated in Fig. 2(a). Therefore, a recent research focus is on adapting trained LDMs for HR image generation without the need for additional training or fine-tuning (*i.e.* training-free manner), which can inherit the strong generation capacities of existing LDMs, especially open-sourced versions like Stable Diffusion.

Existing training-free approaches for HR image generation can be roughly categorized into three types: sliding window-based, parameter rectification-based, and progressive upsampling-based. Sliding window-based methods first divide the HR image into several overlapping patches and use sliding window strategies to perform denoising (Bar-Tal et al., 2023; Haji-Ali et al., 2023; Lee et al., 2023). However, these methods could result in repeated structures and contents due to the lack of communication between windows; see Fig. 2(b). Parameter rectification-based methods attempt to correct models' parameters for better structural consistency through the entropy of attention maps, signal-to-noise ratio, and dilation rates of the convolution layers (Jin et al., 2024; Hwang et al., 2024; He et al., 2023). Though efficient, they often lead to the degradation of texture details; see Fig. 2(c). Unlike the two types mentioned above, progressive upscaling-based methods are to iteratively upsample the image resolution, which maintains better structural consistency and shows state-of-the-art (SOTA) performance (Du et al., 2024; Lin et al., 2024). Unfortunately, these methods require fully repeating the denoising process multiple times, leading to an unaffordable computational burden; *e.g.*, AccDiffusion takes 26 minutes to generate a $4096 \times 4096$ image. In addition, their upsampling operation in the latent space may introduce artifacts; see Fig. 2(d). To sum up, existing methods fail to ensure the fast, high-quality HR image generation.

In this paper, we propose the attentive and progressive LDM, termed AP-LDM, a novel, training-free framework aimed at enhancing HR image quality while speeding up the generation process. Specifically, AP-LDM decomposes the denoising process of LDMs into two stages: (**i**) attentive training-resolution (TR) denoising, and (**ii**) progressive HR denoising. The first stage aims to generate a latent representation of a high-quality image at the training resolution through the proposed attentive guidance, which is implemented via a novel parameter-free self-attention mechanism to improve structural consistency. The second stage aims to progressively upsample the resolution in the pixel space rather than latent space, which can alleviate the severe artifacts caused by the latent space upsampling. By leveraging the effective initialization from the first stage, AP-LDM can perform denoising in the second stage with significantly fewer steps, enhancing the overall efficiency with $5\times$ speedup. Extensive experimental results demonstrate the effectiveness and efficiency of AP-LDM in generating HR images over the state-of-the-art baselines.

**Contributions.** The contributions of this work are summarized as follows. (**i**) We propose AP-LDM, a novel, training-free framework aimed at enhancing the HR high-quality generation while accelerating the generation process. (**ii**) We propose attentive guidance, which can utilize a novel parameter-free self-attention to improve the structural consistency of the latent representation towards high-quality images at the training resolution. (**iii**) We propose progressively upsampling the resolution of latent representation in the pixel space, which can alleviate the artifacts caused by the latent space upsampling. (**iv**) Extensive experimental results demonstrate that the proposed AP-LDM significantly outperforms the SOTA models in terms of image quality and inference time, emphasizing its great potential for real-world applications.

## 2 RELATED WORK

**HR image generation with super-resolution.** An intuitive approach to generating HR images is to first use a pre-trained LDM to generate TR images and then apply a super-resolution model to perform upsampling (Wang et al., 2023; Zhang et al., 2021; Liang et al., 2021; Luo et al., 2024; Wang & Zhang, 2024). Although one can obtain structurally consistent HR images in this way, super-resolution models are primarily focused on enlarging the image, and shown to be unable to produce the details that users expect in HR images (Du et al., 2024; Lin et al., 2024).

**HR image generation with additional training.** Existing additional training methods either fine-tune existing LDMs with higher-resolution images (Hoogeboom et al., 2023; Zheng et al., 2024; Guo et al., 2024) or train cascaded diffusion models to gradually synthesize higher-resolution images (Teng et al., 2023; Ho et al., 2022). Though effective, these methods require expensive training resources that are unaffordable for regular users.

**HR image generation in training-free manner.** Current training-free methods can be roughly classified into three categories: sliding window-based, parameter rectification-based, and progressive upsampling-based methods. Sliding window-based methods consider spatially splitting HR image generation (Bar-Tal et al., 2023; Haji-Ali et al., 2023; Lee et al., 2023). Specifically, they partition an HR image into several patches with overlap, and then denoise each patch. However, due to the lack of communication between windows, these methods result in structural disarray and content duplication. While enlarging the overlaps of the windows mitigates this issue, it can result in unbearable computational costs. For the parameter rectification-based methods, some researchers discovered that the collapse of HR image generation is due to the mismatches between higher resolutions and the model's parameters (Jin et al., 2024; Hwang et al., 2024; He et al., 2023). These methods attempt to eliminate the mismatches by rectifying the parameters such as the dilation rates of some convolutional layers. While mitigating the structural inconsistency, they often lead to the degradation of image details. Different from the aforementioned two types, the progressive upsampling-based methods show SOTA performance in some recent studies (Du et al., 2024; Lin et al., 2024). Though promising, they require fully repeating the denoising process multiple times, which incurs unbearable computational overhead. Additionally, these methods perform upsampling in the latent space, which may introduce artifacts.

These methods aforementioned fail to improve the quality of HR images and computational efficiency at the same time. In contrast to them, AP-LDM aims to enhance both HR images quality and the generation speed towards the real-world applications.

## 3 METHOD

### 3.1 OVERVIEW OF AP-LDM

Fig. 3 presents the overview of AP-LDM, which can adapt a pre-trained LDM to generate HR images without further training. Formally, a pre-trained LDM utilizes a denoising U-Net model $\mathcal{F}$ to iteratively denoise the latent representation of size $h \times w \times c$, which is then converted back to the pixel space for final image generation through the decoder $\mathcal{D}$ of a variational autoencoder (VAE). We note that the initial latent representation is sampled from a Gaussian distribution $\epsilon \sim \mathcal{N}(0, \boldsymbol{I})$, and for the inference the encoder $\mathcal{E}$ of VAE is not involved.

Our AP-LDM extends the pre-trained LDMs for higher-resolution image generation in a training-free manner; *i.e.*, $\mathcal{E}$, $\mathcal{D}$ and $\mathcal{F}$ are fixed. AP-LDM achieves this by decomposing the standard denoising process in the latent space into two stages: (**i**) attentive training-resolution (TR) denoising, and (**ii**) progressive high-resolution (HR) denoising. In the first stage, AP-LDM aims to generate a latent representation of a higher-quality TR image through the proposed attentive guidance. The attentive guidance is implemented as linearly combining the novel parameter-free self-attention (PFSA) and the original latent representation to improve the structural consistency. In the second stage, AP-LDM uses the latent representation provided by the first stage as a better initialization, and iteratively obtains higher-resolution images via the pixel space upsampling and diffusion-denoising refinement.

We detail the attentive TR denoising in §3.2, followed by the progressive HR denoising in §3.3.

Figure 3: **Overview of AP-LDM**. AP-LDM divides the denoising process of a pre-trained LDM into two stages. The first stage leverages the introduced attentive guidance to enhance the structural consistency by utilizing a novel parameter-free self-attention mechanism (PFSA). The second stage iteratively upsamples the latent representation in pixel space to eliminate artifacts.

## 3.2 ATTENTIVE TRAINING-RESOLUTION DENOISING

**Motivation.**   Enhancing the structural consistency helps improve image quality (Si et al., 2024). However, it is challenging to do this in a training-free manner. We observe that the self-attention mechanism presents powerful global spatial modeling capability (Vaswani et al., 2017; Han et al., 2021; Dosovitskiy et al., 2020; Liu et al., 2021), and this capability is parameter-agnostic. It is determined by the paradigm of global similarity calculation inherent to the self-attention mechanism (Vaswani et al., 2017; Zhou et al., 2024). These insights motivate us to consider designing a novel parameter-free self-attention mechanism to elegantly enhance the global structural consistency of the latent representation.

**Denoising with attentive guidance.**   To improve the structural consistency of the latent representation at the training resolution $z \in \mathbb{R}^{h \times w \times c}$, we propose a simple yet effective parameter-free self-attention mechanism for attentive guidance, termed PFSA, formulated as:

$$\text{PFSA}(z) = \text{Flatten}^{-1} \left( \text{Softmax} \left( \frac{\text{Flatten}(z) \cdot \text{Flatten}(z)^{\text{T}}}{\lambda} \right) \cdot \text{Flatten}(z) \right), \quad (1)$$

where the operation $\text{Flatten}$ reshapes the latent representation into shape $(hw) \times c$ and $\text{Flatten}^{-1}$ reshapes it back; $\lambda$ is the scaling factor, with a default value of $\lambda = \sqrt{c}$.

However, we empirically observe that directly using the PFSA in Eq. (1) to improve the structural consistency of the latent representation could lead to unstable denoising. Therefore, we propose linearly combining the outputs of PFSA and the original latent representation as attentive guidance, which is formulated as:

$$\tilde{z} = \gamma \text{PFSA}(z) + (1 - \gamma) z, \quad (2)$$

where $\tilde{z}$ is the structurally enhanced latent representation and $\gamma$ is the guidance scale.

As shown in Fig. 3, we append the attentive guidance in Eq. (2) to denoising U-net model $\mathcal{F}$ and repeat the denoising process for a total of $T_0$ times for the first stage. We note that the denoising process starts from step $T_0$ to 0, and the final output of the first stage is denoted as $z_0^{(0)}$.

**Adaptive guidance scale.**   Considering that the latent representation is mostly non-semantic noise in the first few steps of denoising, we delay $k$ steps in introducing attentive guidance. Moreover, during the denoising process, the image structure is generated first, followed by local details (Yu et al., 2023; Teng et al., 2023). Therefore, we primarily employ attentive guidance in the early to mid-steps of denoising to focus on enhancing the structural consistency of the latent representation. Specifically, we introduce the adaptive guidance scale $\gamma_t$ by applying a decay to a given guidance scale $\gamma$, formulated as:

$$\gamma_t = \begin{cases} \gamma \left[ \frac{\cos\left( \frac{T_0 - k - t}{T_0 - k} \pi \right) + 1}{2} \right]^{\beta} & \text{if } t \leq T_0 - k, \\ 0 & \text{otherwise,} \end{cases} \quad (3)$$

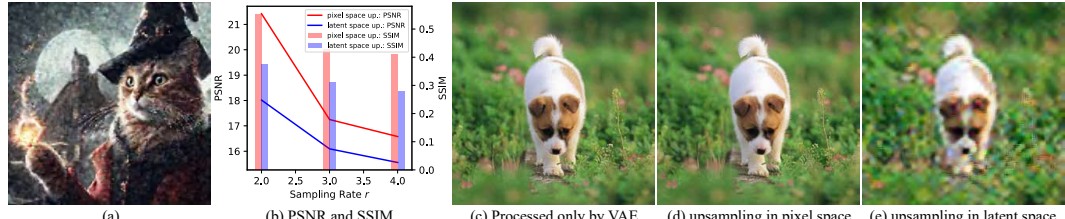

Figure 4: (a) AP-LDM generation with latent space upsampling leads to severe artifacts. (b) Quantitative analysis of PSNR and SSIM. (c) to (e): Qualitative comparisons of VAE-only process and upsampling in different spaces.

where $\beta$ is the decay factor. In practice, considering that $k$ depends on $T_0$ for different resolutions, we use a delay rate $\eta_1 = \frac{k}{T_0}$ to control the number of steps for delaying attentive guidance.

### 3.3 PROGRESSIVE HIGH-RESOLUTION DENOISING

**Motivation.** Fig. 2(a) shows that pre-trained LDMs still retain some ability to generate high-frequency information when directly used to synthesize HR images, although they exhibit structural disarray. Therefore, intuitively, we can utilize the latent representation produced by the first stage as a structural initialization, and generate the HR images through the "upsample-diffuse-denoise" iteration in the latent space. However, this pipeline leads to severe artifacts, as shown in Fig. 4(a). We speculate that this is due to *the upsampling of latent representations in the latent space*.

**Pilot study.** To examine this hypothesis, we conduct the following experiments. Specifically, we randomly select 10k images from ImageNet (Deng et al., 2009) to create an image set $\mathcal{P}$. For each image $\boldsymbol{x} \in \mathcal{P}$, we perform the following operations to obtain three additional image sets: (**i**) $\hat{\boldsymbol{x}} = \mathcal{D} \circ \mathcal{E}(\boldsymbol{x})$, which use VAE to obtain the reconstructed image set $\mathcal{P}_{\text{ref}}$; (**ii**) $\hat{\boldsymbol{x}} = \text{up} \circ \mathcal{D} \circ \mathcal{E} \circ \text{down}(\boldsymbol{x})$, which performs upsampling in pixel space to obtain the image set $\mathcal{P}_{\text{pix}}$; and (**iii**) $\hat{\boldsymbol{x}} = \mathcal{D} \circ \text{up} \circ \mathcal{E} \circ \text{down}(\boldsymbol{x})$, which performs upsampling in latent space to obtain the image set $\mathcal{P}_{\text{lat}}$. Both upsampling up and downsampling down are performed using bicubic interpolation. Fig. 4(b) reports the quantitative results, where $r$ represents the upsampling or downsampling rate. We calculate the PSNR and SSIM for pixel space upsampling set $\mathcal{P}_{\text{pix}}$ and latent space upsampling set $\mathcal{P}_{\text{lat}}$ with respective to the reference set $\mathcal{P}_{\text{ref}}$. It can be clearly observed that the latent space upsampling leads to a significant performance decline compared to pixel space upsampling. Fig. 4(c) to (e) shows upsampling in the pixel space produces images close to the reference while upsampling in latent space leads to severe artifacts and detail loss.

**Progressive HR denoising with pixel space upsampling.** Based on the above conclusion, we propose performing upsampling in the pixel space rather than latent space and utilize diffusion and denoising to refine the upsampled higher-resolution image. Specifically, the second stage consists of $n$ sub-stages to progressively upsample the training-resolution to target resolution, each corresponding to one upsampling operation. For $i$-th sub-stage, $i = 1, \ldots, n$, we prepend an upsample and diffuse operation before the denoising process, which can be defined as:

$$\boldsymbol{z}_{T_i}^{(i)} = \sqrt{\bar{\alpha}_{T_i}} \hat{\boldsymbol{z}}_0^{(i-1)} + \sqrt{1 - \bar{\alpha}_{T_i}} \boldsymbol{\epsilon}, \quad \text{where} \quad \hat{\boldsymbol{z}}_0^{(i-1)} = \mathcal{E}(\mathcal{U}(\mathcal{D}(\boldsymbol{z}_0^{(i-1)}))), \tag{4}$$

where $\bar{\alpha}_{T_i}$ is the noise schedule hyper-parameter of the $T_i$-th diffusion time step, $\boldsymbol{z}_0^{(i-1)}$ is the output of the $(i-1)$-th sub-stage; we use $\boldsymbol{z}_0^{(0)}$ to denote the output from the first stage. Then, $\mathcal{F}$ is used to iteratively denoise $\boldsymbol{z}_{T_i}^{(i)}$ from time step $T_i$ to obtain $\boldsymbol{z}_0^{(i)}$. After completing all sub-stages, we obtain $z_0^{(n)}$, which is then decoded to produce the final output $\boldsymbol{x}^{(n)} = \mathcal{D}(\boldsymbol{z}_0^{(n)})$.

We empirically found that generating higher-resolution images requires more sub-stages. Additionally, when refining images using diffusion and denoising, higher resolutions demand larger time steps (Teng et al., 2023). In practice, for flexibility, AP-LDM allows users to customize the number of sub-stages $n$, and the diffusion time steps $T_i$ for each sub-stage by a pre-specified variable-length progressive scheduler $\eta_2 = \left[\frac{T_1}{T_0}, \frac{T_2}{T_0}, \ldots, \frac{T_n}{T_0}\right]$, whose length is $n$. The elements of $\eta_2$ represent the denoising steps of each sub-stage, normalized by $T_0$.

## 4 EXPERIMENTS

### 4.1 IMPLEMENTATION DETAILS

**Experimental settings.** We use SDXL (Podell et al., 2023) as the pre-trained LDM and conduct inference using two NVIDIA 4090 GPUs. To ensure consistency when testing inference speed, we use a single NVIDIA 3090 GPU, aligning with other methods. We randomly sample 33k images from the SAM (Kirillov et al., 2023) dataset as the benchmark. Following the released code from DemoFusion (Du et al., 2024), we use the EulerDiscreteScheduler (Karras et al., 2022) setting $T_0 = 50$ and the classifier-free guidance (Ho & Salimans, 2022) scale to 7.5. Pixel space upsampling is performed using bicubic interpolation, and the decay factor $\beta$ is fixed at 3.

**Evaluation metrics.** The widely recognized metrics FID (Heusel et al., 2017), IS (Salimans et al., 2016), and CLIP Score (Radford et al., 2021) are used to evaluate model performance. Additionally, since calculating FID and IS requires resizing images to $299 \times 299$, which may not be suitable for evaluating HR images, we are inspired by (Du et al., 2024; Lin et al., 2024) to perform ten $1024 \times 1024$ window crops on each image to calculate $\text{FID}_c$ and $\text{IS}_c$.

### 4.2 QUANTITATIVE RESULTS

We compare AP-LDM with the following models: (1) SDXL (Podell et al., 2023); (2) MultiDiffusion (Bar-Tal et al., 2023); (3) ScaleCrafter (He et al., 2023); (4) DemoFusion (Du et al., 2024); (5) Upsample Guidance (UG) (Hwang et al., 2024); (6) AccDiffusion (Lin et al., 2024). For fair comparisons, we disabled the FreeU trick (Si et al., 2024) in all experiments.

Table 1: **Quantitative comparison results**. The best results are marked in **bold**, and the second best results are marked by underline.

| Method | $2048 \times 2048$ | | | | | $2048 \times 4096$ | | | | | $4096 \times 2048$ | | | | | $4096 \times 4096$ | | | | |
|---|---|---|---|---|---|---|---|---|---|---|---|---|---|---|---|---|---|---|---|---|
| | FID | IS | FID$_c$ | IS$_c$ | CLIP | FID | IS | FID$_c$ | IS$_c$ | CLIP | FID | IS | FID$_c$ | IS$_c$ | CLIP | FID | IS | FID$_c$ | IS$_c$ | CLIP |
| SDXL | 99.9 | 14.2 | 80.0 | 16.9 | 25.0 | 149.9 | 9.5 | 106.3 | 12.0 | 24.4 | 173.1 | 9.1 | 108.5 | 11.5 | 23.9 | 191.4 | 8.3 | 114.1 | 12.4 | 22.9 |
| MultiDiff. | 98.8 | 14.5 | 67.9 | 17.1 | 24.6 | 125.8 | 9.6 | 71.9 | 15.7 | 24.6 | 149.0 | 9.0 | 70.5 | 14.4 | 24.4 | 168.4 | 6.5 | 76.6 | 14.4 | 23.1 |
| ScaleCrafter | 98.2 | 14.2 | 89.7 | 13.3 | 25.4 | 161.9 | 10.0 | 154.3 | 7.5 | 23.3 | 175.1 | 9.7 | 167.3 | 8.0 | 21.6 | 164.5 | 9.4 | 170.1 | 7.3 | 22.3 |
| UG | 82.2 | 17.6 | 65.8 | 14.6 | 25.5 | 155.7 | 8.2 | 165.0 | 6.6 | 21.7 | 185.3 | 6.8 | 175.7 | 6.2 | 20.5 | 187.3 | 7.0 | 197.6 | 6.3 | 21.8 |
| DemoFusion | 72.3 | 21.6 | 53.5 | 19.1 | 25.2 | 96.3 | 17.7 | 62.3 | 15.0 | 25.0 | 99.6 | 16.4 | 61.9 | 14.7 | 24.4 | 101.4 | 20.7 | 63.5 | 13.5 | 24.7 |
| AccDiff. | 71.6 | 21.0 | 52.7 | 17.0 | 25.1 | 95.5 | 16.4 | 62.9 | 11.1 | 24.5 | 102.2 | 15.2 | 65.4 | 11.5 | 24.2 | 103.2 | 20.1 | 65.9 | 13.3 | 24.6 |
| AP-LDM | 66.0 | 21.0 | 47.4 | 17.5 | 25.1 | 89.0 | 20.3 | 56.0 | 19.0 | 25.0 | 93.2 | 19.5 | 56.9 | 16.5 | 24.9 | 90.6 | 21.1 | 59.0 | 14.8 | 24.6 |

We report the performance of all methods on four different resolutions (Height $\times$ Width): $4096 \times 4096$, $4096 \times 2048$, $2048 \times 4096$, and $2048 \times 2048$. Considering that the generation time for HR images far exceeds that for low-resolution images, we used 2k prompts at the resolution of $2048 \times 2048$, and 1k prompts for resolutions greater $2048 \times 2048$. For all resolutions, we set $\gamma = 0.004$ and $\eta_1 = 0.06$ for AP-LDM. Given that the $4096 \times 4096$ resolution is significantly larger than other resolutions, we set $\eta_2 = [0.1, 0.2]$ (*i.e.*, $T_0 = 50$, $T_1 = 5$, $T_2 = 10$) for $4096 \times 4096$, and $\eta_2 = [0.2]$ (*i.e.*, $T_0 = 50$ and $T_1 = 10$) for other resolutions. When generating images with an aspect ratio of $r'$, we reshape the initially sampled Gaussian noise $\epsilon$ in the first stage to match $r'$. This process keeps the number of tokens in $\epsilon$ unchanged, preventing drastic fluctuations in the entropy of the attention maps in the transformer (Jin et al., 2024) leading to higher-quality images.

Table 1 manifests that AP-LDM significantly outperforms previous SOTA models, AccDiffusion and DemoFusion. This indicates that AP-LDM generates images with higher quality. Notably, Table 2 indicates that AP-LDM demonstrates remarkable advantage in inference speed compared to the SOTA models. On a consumer-grade 3090 GPU, AP-LDM requires only about one-fifth of the inference time needed by SOTA models such as DemoFusion and AccDiffusion.

Table 2: **Model inference time**. The best results are marked in **bold**. Unit of Time: minute.

| Resolutions | SDXL | MultiDiff. | ScaleCrafter | UG | DemoFusion | AccDiff. | AP-LDM |
|---|---|---|---|---|---|---|---|
| $2048 \times 2048$ | 1.0 | 3.0 | 1.0 | 1.8 | 3.0 | 3.0 | **0.6** |
| $2048 \times 4096$ | 3.0 | 6.0 | 6.0 | 4.0 | 11.0 | 12.7 | **2.0** |
| $4096 \times 4096$ | 8.0 | 15.0 | 19.0 | 11.1 | 25.0 | 26.0 | **5.7** |

### 4.3 QUALITATIVE RESULTS

In Fig. 5, AP-LDM is qualitatively compared with AccDiffusion, DemoFusion, and MultiDiffusion. MultiDiffusion fails to maintain global semantic consistency. As indicated by the red boxes, Demo-Fusion and AccDiffusion tend to result in chaotic content repetition and severe artifacts, which we speculate are caused by upsampling in the latent space (as analyzed in §3.3). In contrast, AP-LDM not only preserves excellent global structural consistency but also synthesizes images with more details. More qualitative comparison results can be found in Appendix A.3.

### 4.4 USER STUDY

We invite 16 volunteers to participate in a double-blind experiment to further evaluate the performance of the models. Each volunteer is required to answer 35 questions. In each question, three images gener-

Table 3: **Results of the user study.**

| Method | Structural Consistency | | Color Abundance | | Detail Richness | |
|---|---|---|---|---|---|---|
| | score ↑ | score* ↑ | score ↑ | score* ↑ | score ↑ | score* ↑ |
| AccDiffusion | 6.28 | 0.88 | 6.78 | 0.60 | 6.18 | 0.53 |
| DemoFusion | 5.99 | 0.59 | 6.69 | 0.51 | 6.18 | 0.53 |
| AP-LDM | **7.42** | **2.02** | **7.64** | **1.45** | **7.41** | **1.76** |

ated by AccDiffusion, DemoFusion, and AP-LDM are presented. The volunteer needs to rate each image from 1 to 10 in terms of structural consistency, color abundance, and detail richness. We calculate the average of their scores. Moreover, to eliminate bias in each volunteer's ratings for each metric in each question, we subtract the minimum value among the three scores given by each volunteer for each metric in each question. The rectified score is denoted as score*. Table 3 shows that AP-LDM surpasses previous SOTA models across all metrics.

## 5 ABLATION STUDY

### 5.1 ATTENTIVE GUIDANCE

In this section, we first conduct ablation experiments on attentive guidance, followed by ablation experiments on the hyper-parameters of attentive guidance.

**Ablation on attentive guidance.** We keep $\eta_2$ unchanged and analyze the effect of attentive guidance through qualitative and quantitative experiments. Table. 4 shows that attentive guidance leads to improvements across various metrics, indicating that using attentive guidance to enhance the consistency of latent encoding results in higher-quality images. The qualitative experiments in Fig. 6 demonstrate that using attentive guidance eliminates image blurriness and enriches the image details. Please refer to Appendix A.2.1 for additional qualitative ablation results.

Table 4: **Ablation on attentive guidance (AG).** The best results are marked in **bold**.

| Method | $2048 \times 2048$ | | | | | $2048 \times 4096$ | | | | | $4096 \times 2048$ | | | | | $4096 \times 4096$ | | | | |
|---|---|---|---|---|---|---|---|---|---|---|---|---|---|---|---|---|---|---|---|---|
| | FID | IS | $FID_c$ | $IS_c$ | CLIP | FID | IS | $FID_c$ | $IS_c$ | CLIP | FID | IS | $FID_c$ | $IS_c$ | CLIP | FID | IS | $FID_c$ | $IS_c$ | CLIP |
| w/o AG | 66.8 | **21.6** | 47.5 | 17.4 | **25.3** | 91.6 | **20.3** | 58.0 | 14.5 | **25.0** | 95.3 | **19.9** | 58.4 | 14.5 | **24.9** | 92.0 | **21.6** | 59.8 | 13.6 | 24.5 |
| w/ AG | **66.0** | 21.0 | **47.4** | **17.5** | 25.1 | **89.0** | 20.3 | **56.0** | **19.0** | **25.0** | **93.2** | 19.5 | **56.9** | **16.5** | **24.9** | **90.6** | 21.1 | **59.0** | **14.8** | **24.6** |

**Ablation on guidance scale $\gamma$.** We fix $\eta_1 = 0.06, \eta_2 = [0.2]$ and then explore the effect of the guidance scale $\gamma$ through both quantitative and qualitative experiments. For the quantitative experiments, we find that $\gamma = 0.004$ performs better. Interestingly, when a larger guidance scale is used, the visual quality of the images can be further enhanced. As shown in Fig. 7, using a larger guidance scale results in richer image details. This allows users to generate images according to their preferences for detail richness and color contrast by adjusting the guidance scale. The setup and results of the quantitative experiments are detailed in Appendix A.2.1.

**Ablation on delay rate $\eta_1$.** We fix $\gamma = 0.004, \eta_2 = [0.2]$ and then investigate the impact of the delay rate $\eta_1$ through both quantitative and qualitative experiments. The quantitative analysis results indicate that better generation results can be achieved when $\eta_1 = 0.06$, indicating that appropriately delaying the effect of attentive guidance contributes to further improving the quality of the images. We conjecture that this is because, at the very beginning of the denoising process, the structural information in the latent encoding has not yet emerged, and thus attentive guidance cannot effectively enhance structural consistency. As shown in Fig. 8, delaying the effect of attentive guidance eliminates some generation errors, further improving image quality. The setup and results of the quantitative experiments are detailed in Appendix A.2.1.

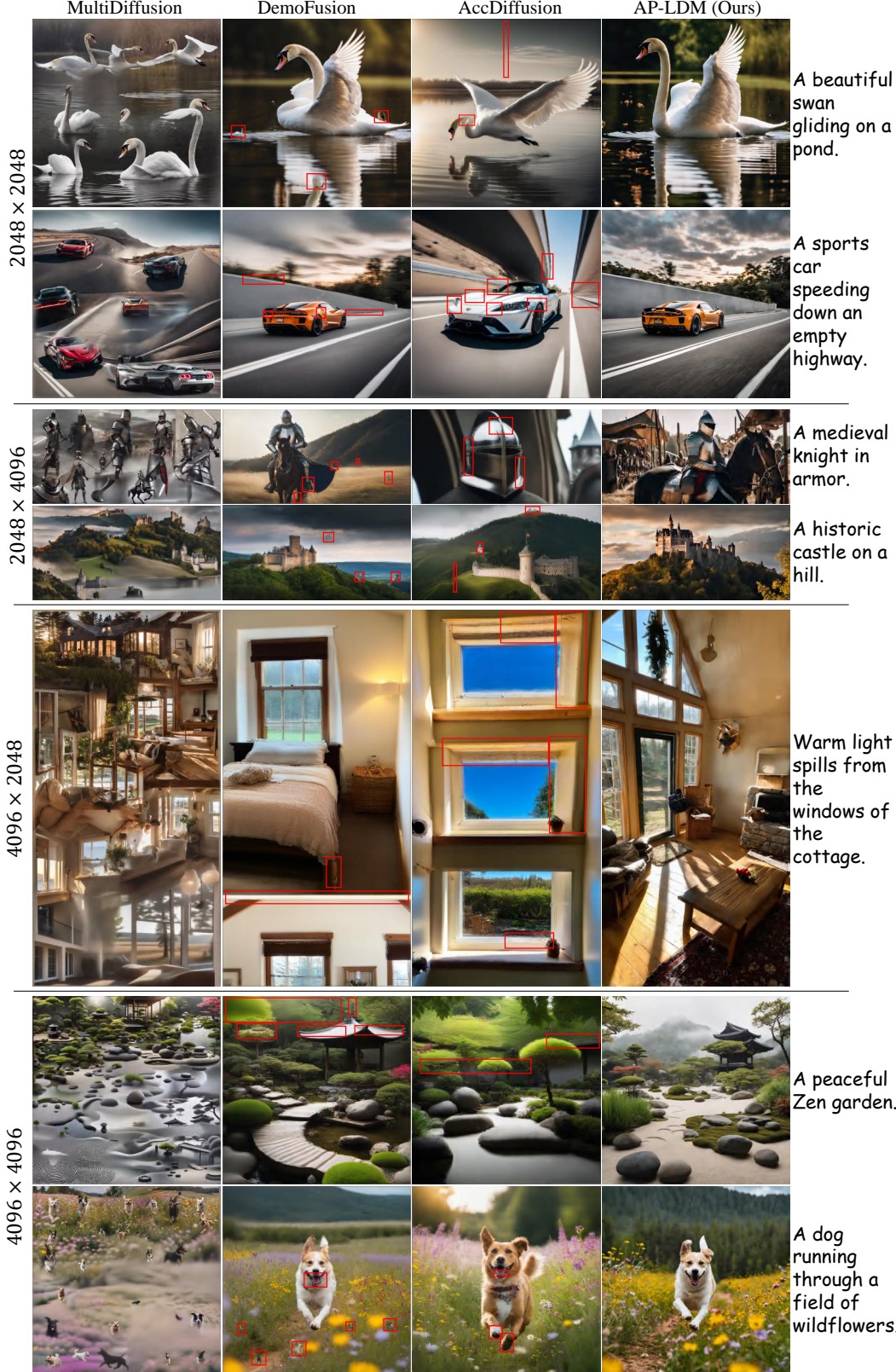

Figure 5: **Qualitative comparison with other baselines.** The prompts used to generate the images are presented on the right. MultiDiffusion fails to maintain global semantic consistency. DemoFusion and AccDiffusion exhibit severe artifacts and content repetition. The red boxes indicate some synthesis errors.

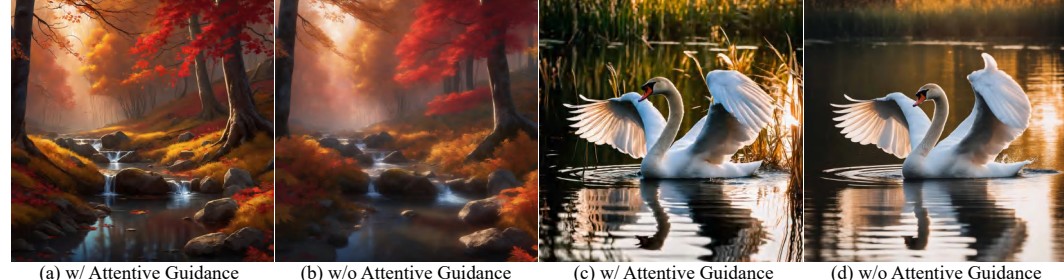

(a) w/ Attentive Guidance    (b) w/o Attentive Guidance    (c) w/ Attentive Guidance    (d) w/o Attentive Guidance

Figure 6: **Generating with and without attentive guidance**. Resolution: $2048 \times 2048$. Zoom-in for a better view.

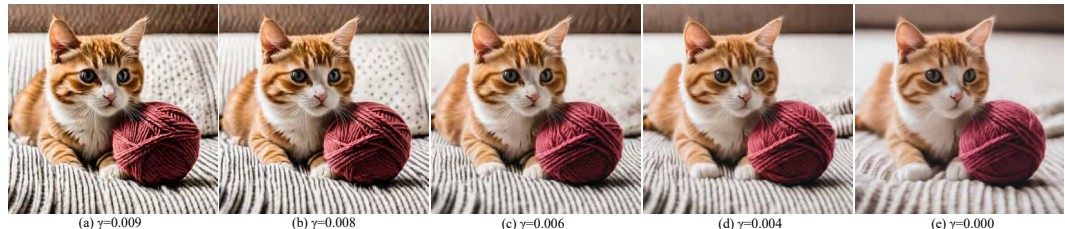

(a) $\gamma=0.009$    (b) $\gamma=0.008$    (c) $\gamma=0.006$    (d) $\gamma=0.004$    (e) $\gamma=0.000$

Figure 7: **Generating images using different guidance scale**. Resolution: $2048 \times 2048$.

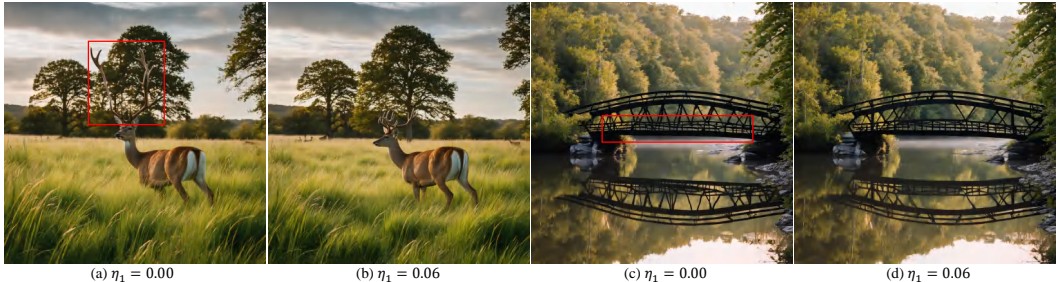

(a) $\eta_1 = 0.00$    (b) $\eta_1 = 0.06$    (c) $\eta_1 = 0.00$    (d) $\eta_1 = 0.06$

Figure 8: **Ablation on delay rate.** Errors indicated by red boxes can be eliminated by delaying attentive guidance. Resolution: $2048 \times 2048$.

**Ablation on the time steps of attentive guidance.**    To explain why attentive guidance needs to be applied during the early to middle steps of denoising, we apply attentive guidance during different denoising steps of the first stage: (a) 47 to 33, (b) 32 to 17, and (c) 16 to 1. Fig. 9 shows that when attentive guidance is applied during the early to middle steps of denoising, the image becomes clearer and more detailed; however, when attentive guidance is applied during the later steps of denoising, it has negligible effect on the generated image. We speculate that this is because diffusion models tend to synthesize structural information first (Teng et al., 2023), and once the structural information is generated, attentive guidance may have a limited impact on structural consistency.

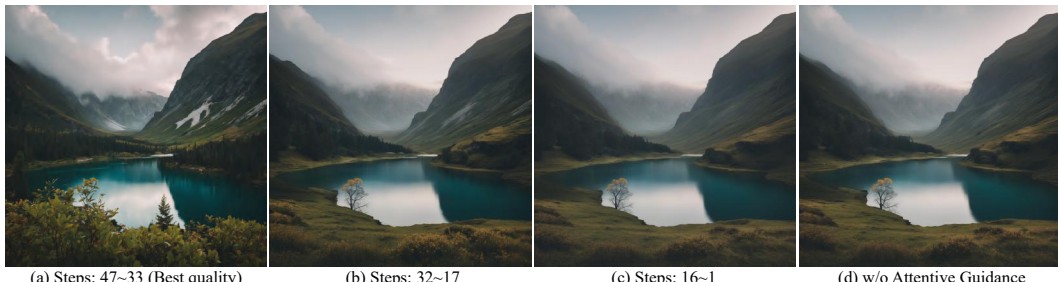

(a) Steps: 47~33 (Best quality)    (b) Steps: 32~17    (c) Steps: 16~1    (d) w/o Attentive Guidance

Figure 9: **Applying attentive guidance at different denoising steps.** Resolution: $2048 \times 2048$.

## 5.2 PROGRESSIVE HIGH-RESOLUTION DENOISING

In this section, we conduct ablation experiments on the progressive scheduler $\eta_2$ in the second stage of AP-LDM. Specifically, we fixed $\gamma = 0, \eta_1 = 0$ and then explore the effect of the progressive scheduler $\eta_2$ through both quantitative and qualitative experiments. Quantitative experimental re-

sults indicate that an excessively large progressive scheduler value may result in a decline in image quality. This can also be observed in Fig. 10. It is evident that a too large progressive scheduler value may lead to structural misalignment and repetition issues observed in pre-trained SDXL. When the progressive scheduler value is sufficiently small, changing it yields similar visual effects. Therefore, we can choose a smaller progressive scheduler value (*e.g.*, 0.2) to accelerate inference. The setup and quantitative results are detailed in Appendix A.2.2.

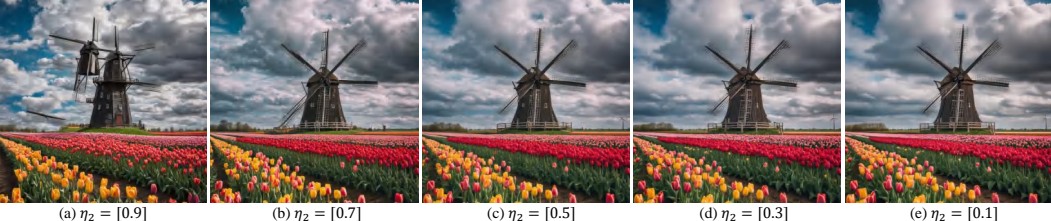

(a) $\eta_2 = [0.9]$     (b) $\eta_2 = [0.7]$     (c) $\eta_2 = [0.5]$     (d) $\eta_2 = [0.3]$     (e) $\eta_2 = [0.1]$

Figure 10: **Generated** $2048 \times 2048$ **images using different** $\eta_2$. (a): When the value of progressive scheduler is too large, the structural repetition issue may reappear. (b) to (e): The visual effects are similar. Therefore, we can use a smaller progressive scheduler value to accelerate inference.

## 6 LIMITATIONS AND FUTURE WORK

AP-LDM exhibits limitations in the following aspects: (**i**) Effectively controlling text in images is challenging, as demonstrated by examples in Fig. 11. This may be due to the inherent limitations of SDXL in generating textual symbols. Text, due to its more regular structure compared to other image content, is difficult to restore by directly enhancing the structural consistency of the latent representation. We speculate that the most reliable approach would be to fine-tune the model specifically on images containing text. (**ii**) When generating ultra-high resolution images, such as $12800 \times 12800$, the second stage of AP-LDM inevitably needs to be decomposed into more sub-stages, which increases the model's inference time.

Developing a low-cost and effective fine-tuning method to correct text generation errors may be a promising direction. Moreover, adapting attentive guidance to other tasks, such as video generation can be an interesting issue.

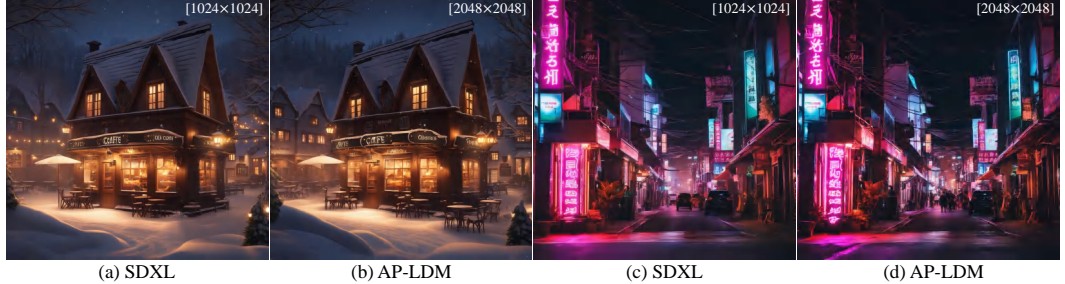

(a) SDXL     (b) AP-LDM     (c) SDXL     (d) AP-LDM

Figure 11: **Limitations of AP-LDM.** The generation results of SDXL at its training resolution and those of AP-LDM at higher resolutions are provided.

## 7 CONCLUSION

In this paper, we propose an effective, efficient, and training-free pipeline named AP-LDM, capable of generating HR images with higher quality while accelerating the generation process. AP-LDM divides the denoising process of an LDM into two stages: (**i**) attentive training-resolution denoising, and (**ii**) progressive high-resolution denoising. The first stage aims to generate a latent representation through the proposed attentive guidance, which enhances the structural consistency by leveraging a novel parameter-free self-attention mechanism. The second stage iteratively performs upsampling in the pixel stage, thus eliminating the artifacts caused by latent space upsampling. Extensive experiments show that our proposed AP-LDM significantly outperforms SOTA models while achieving $5\times$ speedup in HR image generation.

## 8 REPRODUCIBILITY STATEMENT

We make the following efforts to ensure the reproducibility of AP-LDM: (**i**) Our code will be made publicly available. (**ii**) We provide implementation details in §3 and Appendix 1.

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

---

**Algorithm 1** AP-LDM Inference Pipeline

---

**Require:** The number of inference time steps of the first stage $T_0$; progressive scheduler $\eta_2$; attentive guidance scale $\gamma$; attentive guidance delay rate $\eta_1$; the decay factor $\beta$; target image size tuple $(H', W')$; the denoising model $\mathcal{F}$; denoising model's training resolution tuple $(H, W)$; VAE encoder $\mathcal{E}$; VAE decoder $\mathcal{D}$; noise scheduler's hyper-parameter list $\bar{\alpha}_{1:T_0}$.

1: **Initialization:**
2: $\quad z_{T_0}^{(0)} = \epsilon \sim \mathcal{N}(0, I)$ $\hfill \triangleright$ Sampling from Standard Gaussian Distribution
3: $\quad n_{\text{stages}} = \text{length}(\eta_2) + 1$ $\hfill \triangleright$ Get the total number of denoising stages
4: $\quad r' = \frac{H'}{W'}$ $\hfill \triangleright$ Keep the aspect ratio and number of pixels unchanged
5: $\quad H^{(0)} = \text{ceil}(\sqrt{H \times W \times r'})$
6: $\quad W^{(0)} = \text{ceil}(\sqrt{\frac{H \times W}{r'}})$
7: $\quad H^{(n)} = H'$
8: $\quad W^{(n)} = W'$
9: $\quad area_{\text{list}} = \text{linspace}(H^{(0)} \times W^{(0)}, H^{(n)} \times W^{(n)}, n_{\text{stages}})$ $\triangleright$ Upsampling according to the number of pixels
10: $\quad H_{\text{list}} = [\text{ceil}(\sqrt{i \times r'}) \quad \text{for} \quad i \quad \text{in} \quad area_{\text{list}}]$ $\hfill \triangleright$ Get the height and width of each stage
11: $\quad W_{\text{list}} = [\text{ceil}(\sqrt{i/r'}) \quad \text{for} \quad i \quad \text{in} \quad area_{\text{list}}]$
12: $\quad k_{\text{denoising}} = [T_0]$ $\hfill \triangleright$ Get the number of denoising steps for each stage
13: $\quad k_{\text{denoising}}.\text{extend}([i \times T_0 \quad \text{for} \quad i \quad \text{in} \quad \eta_2])$
14: $\quad k = T_0 \times \eta_1$ $\hfill \triangleright$ Obtain the number of delay steps
15: $\quad \gamma_{\text{list}} = [\gamma(\frac{cos(\frac{T-k-i}{T-k}\pi)+1}{2})^{\beta} \quad \text{for} \quad i = 1, ..., T - k]$ $\hfill \triangleright$ Obtain the guidance scale for each step
16: **Denoising:**
17: **for** $s = 0, \ldots, n_{\text{stages}} - 1$ **do:**
18: $\quad n_{\text{steps}} \leftarrow k_{\text{denoising}}[s]$
19: $\quad$ **if** $s \geq 1$ **then:**
20: $\quad\quad x^{(s)} \leftarrow \text{upsample}(x^{(s-1)}, H_{\text{list}}[s], W_{\text{list}}[s])$ $\hfill \triangleright$ Upsampling in pixel space
21: $\quad\quad z_0^{(s)} \leftarrow \mathcal{E}(x^{(s)})$
22: $\quad\quad z_{n_{\text{steps}}}^{(s)} \sim \mathcal{N}(\sqrt{\bar{\alpha}[n_{\text{steps}}]}z_0^{(s)}, (1 - \bar{\alpha}[n_{\text{steps}}])I)$
23: $\quad$ **end if**
24: $\quad$ **for** $t = n_{\text{steps}} - 1, \ldots, 0$ **do:**
25: $\quad\quad z_t^{(s)} \leftarrow \mathcal{F}(z_{t+1}^{(s)}, t + 1)$ $\hfill \triangleright$ Denoising
26: $\quad\quad$ **if** $s == 0$ and $t \leq T - 1 - k$ **then:**
27: $\quad\quad\quad z_t^{(s)} \leftarrow \gamma_{\text{list}}[t]\text{PFSA}(z_t^{(s)}) + (1 - \gamma_{\text{list}}[t])z_t^{(s)}$ $\hfill \triangleright$ Attentive Guidance
28: $\quad\quad$ **end if**
29: $\quad$ **end for**
30: $\quad x^{(s)} \leftarrow \mathcal{D}(z_0^{(s)})$ $\hfill \triangleright$ Obtain the pixel space image
31: **end for**

---

# A APPENDIX

## A.1 AP-LDM ALGORITHM

The implementation details of AP-LDM can be found in Algorithm 1, and further information is available in our code repository.

## A.2 FURTHER RESULTS OF ABLATION STUDIES

### A.2.1 ATTENTIVE GUIDANCE

**Quantitative analysis of guidance scale.** We sampled 1k prompts, fixed $\eta_1 = 0.06, \eta_2 = [0.2]$ and performed ablation studies for guidance scale $\gamma$. The quantitative results are shown in Table 5. Considering all metrics, we find that $\gamma = 0.004$ achieved better quantitative results.

**Quantitative analysis of delay rate.** We sampled 1k prompts, fixed $\gamma = 0.004, \eta_2 = [0.2]$ and performed ablation studies for delay rate $\eta_1$. Table 6 presents the experimental results, indicating that better results can be achieved when $\eta_1 = 0.06$. This means that appropriately delaying the effect of attentive guidance can further enhance the quality of the generated images.

**Further qualitative analysis of attentive guidance.** Fig. 12 provides additional qualitative ablation results on attentive guidance. Individual preferences for contrast, color vividness, and detail

| Method | 1024 × 1024 | | | | | 1600 × 1600 | | | | | 2048 × 2048 | | | | |
|---|---|---|---|---|---|---|---|---|---|---|---|---|---|---|---|
| | FID ↓ | IS ↑ | FID$_c$ ↓ | IS$_c$ ↑ | CLIP ↑ | FID ↓ | IS ↑ | FID$_c$ ↓ | IS$_c$ ↑ | CLIP ↑ | FID ↓ | IS ↑ | FID$_c$ ↓ | IS$_c$ ↑ | CLIP ↑ |
| $\gamma = 0.000$ | 90.85 | 58.18 | 21.21 | **17.69** | 25.09 | 90.91 | 54.74 | **21.45** | 15.41 | 24.93 | 91.78 | 59.08 | 21.57 | 17.36 | 24.86 |
| $\gamma = 0.001$ | 90.50 | 58.04 | **21.34** | 16.76 | 25.08 | 91.17 | 54.31 | 21.19 | 15.47 | 24.93 | 91.40 | 58.75 | **21.87** | 15.85 | 24.86 |
| $\gamma = 0.002$ | 89.82 | 57.54 | 21.28 | 17.04 | 25.08 | 90.39 | **53.71** | 21.26 | 15.00 | 24.97 | 90.81 | 58.34 | 21.45 | 17.16 | 24.90 |
| $\gamma = 0.003$ | 90.10 | 57.08 | 20.80 | 16.61 | 25.08 | 90.56 | 53.95 | 21.35 | 15.46 | 24.98 | 90.87 | 58.40 | 21.47 | **17.60** | 24.92 |
| $\gamma = 0.004$ | **89.40** | 56.64 | 20.96 | 16.63 | 25.09 | **89.91** | 54.23 | 20.91 | 15.54 | 25.01 | **90.11** | 58.11 | 21.18 | 16.78 | 24.94s |
| $\gamma = 0.005$ | 90.17 | 57.50 | 20.89 | 16.34 | 25.12 | 90.24 | 55.19 | 20.67 | 15.21 | 25.02 | 90.46 | 58.91 | 20.79 | 16.87 | 24.97 |
| $\gamma = 0.006$ | 89.79 | 58.18 | 20.33 | 15.93 | 25.16 | 90.36 | 56.71 | 20.33 | 14.59 | 25.06 | 90.32 | 59.86 | 20.37 | 16.12 | 25.00 |
| $\gamma = 0.007$ | 90.42 | 60.29 | 20.07 | 16.20 | 25.21 | 90.91 | 59.35 | 20.36 | 14.16 | 25.12 | 90.86 | 61.81 | 20.14 | 15.70 | 25.06 |
| $\gamma = 0.008$ | 91.64 | 63.63 | 19.66 | 14.25 | **25.25** | 91.98 | 63.93 | 19.13 | 13.71 | 25.13 | 92.16 | 64.82 | 19.59 | 14.24 | 25.08 |
| $\gamma = 0.009$ | 94.29 | 67.87 | 19.15 | 13.00 | 25.25 | 94.38 | 70.21 | 19.45 | 12.12 | 25.16 | 94.39 | 68.84 | 19.22 | 13.63 | 25.12 |

Table 5: **Quantitative ablation experiments on the guidance scale** $\gamma$. The best results are marked in **bold**, and the second best results are marked by underline.

| Method | 1024 × 1024 | | | | | 1600 × 1600 | | | | | 2048 × 2048 | | | | |
|---|---|---|---|---|---|---|---|---|---|---|---|---|---|---|---|
| | FID ↓ | IS ↑ | FID$_c$ ↓ | IS$_c$ ↑ | CLIP ↑ | FID ↓ | IS ↑ | FID$_c$ ↓ | IS$_c$ ↑ | CLIP ↑ | FID ↓ | IS ↑ | FID$_c$ ↓ | IS$_c$ ↑ | CLIP ↑ |
| $\eta_1 = 0.00$ | 89.98 | 58.29 | 20.74 | 16.48 | 25.06 | 90.89 | 55.54 | 21.00 | 14.42 | 24.98 | 90.75 | 59.41 | 20.54 | 16.99 | 24.91 |
| $\eta_1 = 0.02$ | 89.96 | 57.67 | 20.99 | **16.87** | 25.05 | 90.76 | 54.77 | 21.08 | 15.35 | 24.95 | 91.78 | 59.08 | **21.57** | **18.16** | 24.86 |
| $\eta_1 = 0.04$ | 89.47 | 57.28 | 20.98 | 16.63 | 25.07 | 90.22 | 54.14 | 20.86 | 15.43 | 24.98 | 90.52 | 58.47 | 20.76 | 17.02 | 24.91 |
| $\eta_1 = 0.06$ | 89.44 | 56.64 | 20.92 | 16.58 | **25.11** | 89.91 | 54.23 | 20.91 | 15.54 | 25.01 | **90.11** | 58.11 | 21.18 | 16.78 | **24.92** |
| $\eta_1 = 0.08$ | 89.95 | 56.97 | 21.05 | 16.76 | 25.09 | 89.87 | 54.10 | 21.22 | 15.65 | 24.98 | 90.74 | 58.45 | 20.99 | 17.06 | **24.92** |
| $\eta_1 = 0.10$ | 89.29 | 56.88 | 21.11 | 16.84 | 25.09 | 89.97 | 53.99 | 21.04 | 15.37 | 24.99 | 90.41 | 58.45 | 20.99 | 17.12 | 24.92 |
| $\eta_1 = 0.12$ | 89.84 | 57.32 | 21.05 | 16.58 | 25.08 | 90.00 | 53.85 | 21.24 | 15.81 | 24.93 | 90.24 | 58.45 | 21.24 | 17.36 | 24.90 |
| $\eta_1 = 0.14$ | 89.85 | 57.12 | 20.91 | 16.40 | 25.09 | 90.06 | 53.83 | 21.33 | 15.62 | 24.99 | 90.69 | 58.25 | 21.17 | 16.74 | 24.91 |
| $\eta_1 = 0.16$ | 90.06 | 57.28 | 21.10 | 16.53 | 25.09 | 90.91 | 54.74 | 21.45 | 15.41 | 24.93 | 90.76 | 58.37 | 20.97 | 16.87 | 24.91 |
| $\eta_1 = 0.18$ | 90.16 | 57.29 | 20.88 | 15.10 | 25.08 | 90.26 | 53.79 | 21.06 | 15.07 | 24.97 | 90.78 | 58.33 | 21.05 | 17.21 | 24.90 |

Table 6: **Quantitative ablation experiments on the delay rate** $\eta_1$. The best results are marked in **bold**, and the second best results are marked by underline.

richness may vary. attentive guidance allows users to adjust parameters such as the guidance scale to synthesize images according to their preferences.

### A.2.2 PROGRESSIVE HIGH-RESOLUTION DENOISING

This section presents the results of quantitative ablation analysis on the progressive scheduler $\eta_2$ in the second stage of AP-LDM. We fixed $\gamma = 0, \eta_1 = 0$, sampled 500 prompts, and generated 1k images to investigate the optimal value of the progressive scheduler. Table 7 presents the quantitative results, indicating that using an excessively large progressive scheduler may lead to a decline in image quality.

| Method | 1600 × 1600 | | | | | 2048 × 2048 | | | | |
|---|---|---|---|---|---|---|---|---|---|---|
| | FID ↓ | IS ↑ | FID$_c$ ↓ | IS$_c$ ↑ | CLIP ↑ | FID ↓ | IS ↑ | FID$_c$ ↓ | IS$_c$ ↑ | CLIP ↑ |
| SDXL | 101.56 | 25.78 | 73.67 | 21.23 | 26.87 | 112.64 | 18.44 | 79.03 | 20.61 | 26.55 |
| $\eta_2 = [0.9]$ | 94.59 | 27.04 | 67.60 | 23.01 | 26.97 | 97.14 | 24.48 | 64.34 | 22.14 | 26.59 |
| $\eta_2 = [0.8]$ | 93.13 | 28.80 | 65.67 | 24.83 | 26.99 | 93.93 | 26.75 | 60.84 | 23.27 | 26.77 |
| $\eta_2 = [0.7]$ | **92.05** | 29.44 | 65.35 | 24.97 | 27.07 | 92.50 | 28.17 | 57.34 | 24.05 | 26.93 |
| $\eta_2 = [0.6]$ | 92.94 | 30.79 | 64.57 | 24.29 | 27.11 | 91.86 | 30.45 | 55.38 | 24.96 | 26.98 |
| $\eta_2 = [0.5]$ | 92.73 | 30.65 | 63.43 | 24.26 | 27.13 | 91.80 | 31.18 | 54.32 | 24.48 | 27.02 |
| $\eta_2 = [0.4]$ | 93.04 | 30.96 | 63.33 | 24.77 | 27.14 | **91.71** | 32.47 | 53.72 | 25.16 | 27.03 |
| $\eta_2 = [0.3]$ | 92.93 | 30.91 | **63.09** | 24.84 | 27.15 | 92.39 | 30.72 | 53.32 | 26.63 | 27.07 |
| $\eta_2 = [0.2]$ | 93.09 | **31.17** | 63.23 | **25.71** | 27.17 | 92.71 | 30.45 | **53.19** | 26.19 | 27.12 |
| $\eta_2 = [0.1]$ | 93.44 | 30.69 | 63.75 | 25.18 | **27.22** | 92.94 | 30.69 | 53.77 | 24.71 | **27.18** |

Table 7: **Quantitative ablation study of the progressive scheduler**. The best results are marked in **bold**, and the second best results are marked by underline.

### A.3 SUPPLEMENTARY QUALITATIVE ANALYSIS

Fig. 13 presents additional qualitative comparison results. MultiDiffusion continues to struggle with maintaining global consistency; as indicated by the red boxes, DemoFusion tends to produce repetitive content, a problem somewhat alleviated in AccDiffusion but not fully resolved. As highlighted by the black boxes, another issue with AccDiffusion is the presence of noticeable streak artifacts in the images.

w/ AG      w/o AG      w/ AG      w/o AG

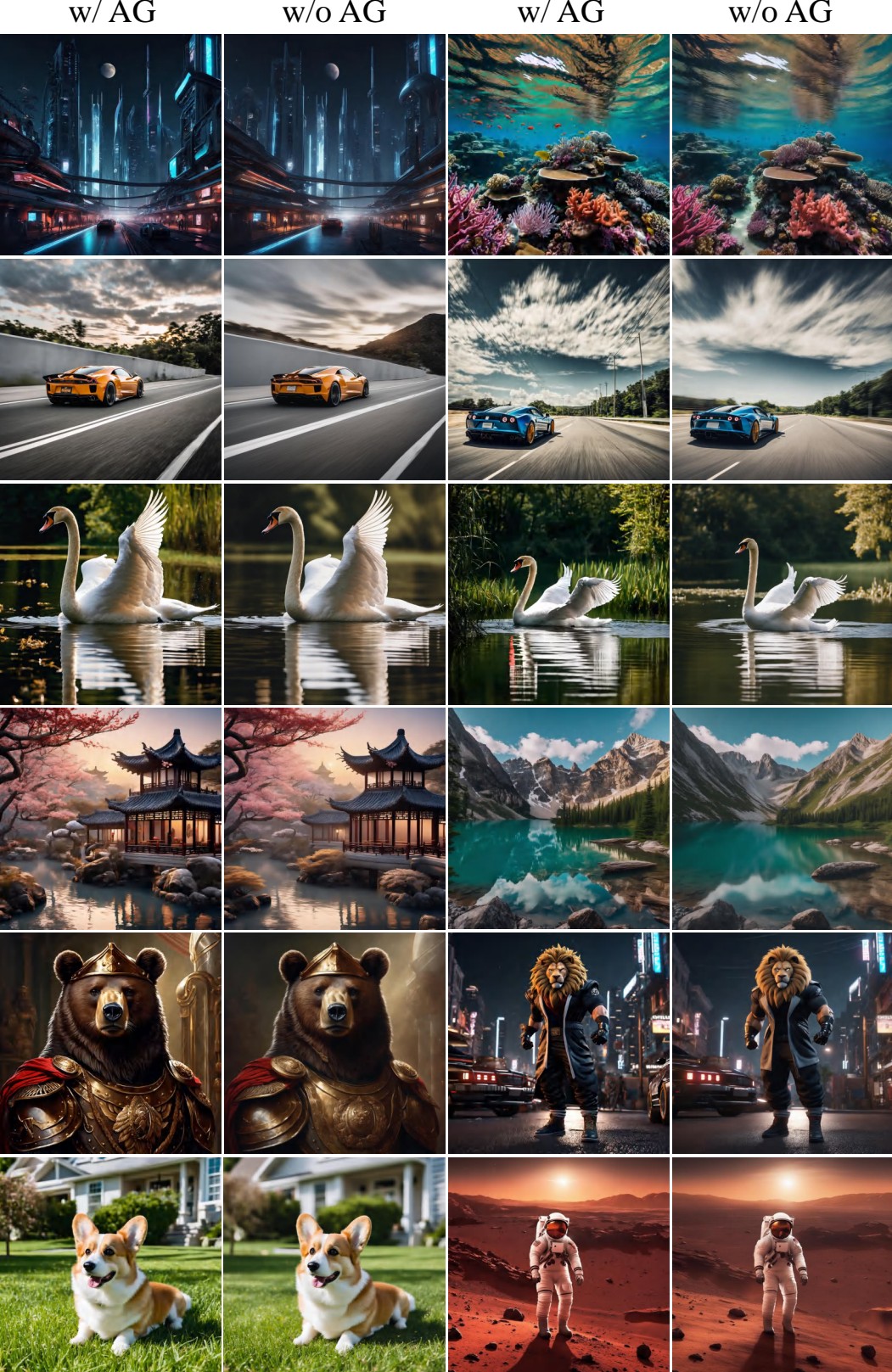

Figure 12: **Further qualitative analysis of attentive guidance (AG).** Using attentive guidance significantly enhances image quality. The details were enriched, for example: the clouds in the sky, ripples on the water, reflections on the lake, and even the expressions in a person's eyes. Best viewed **ZOOMED-IN**.

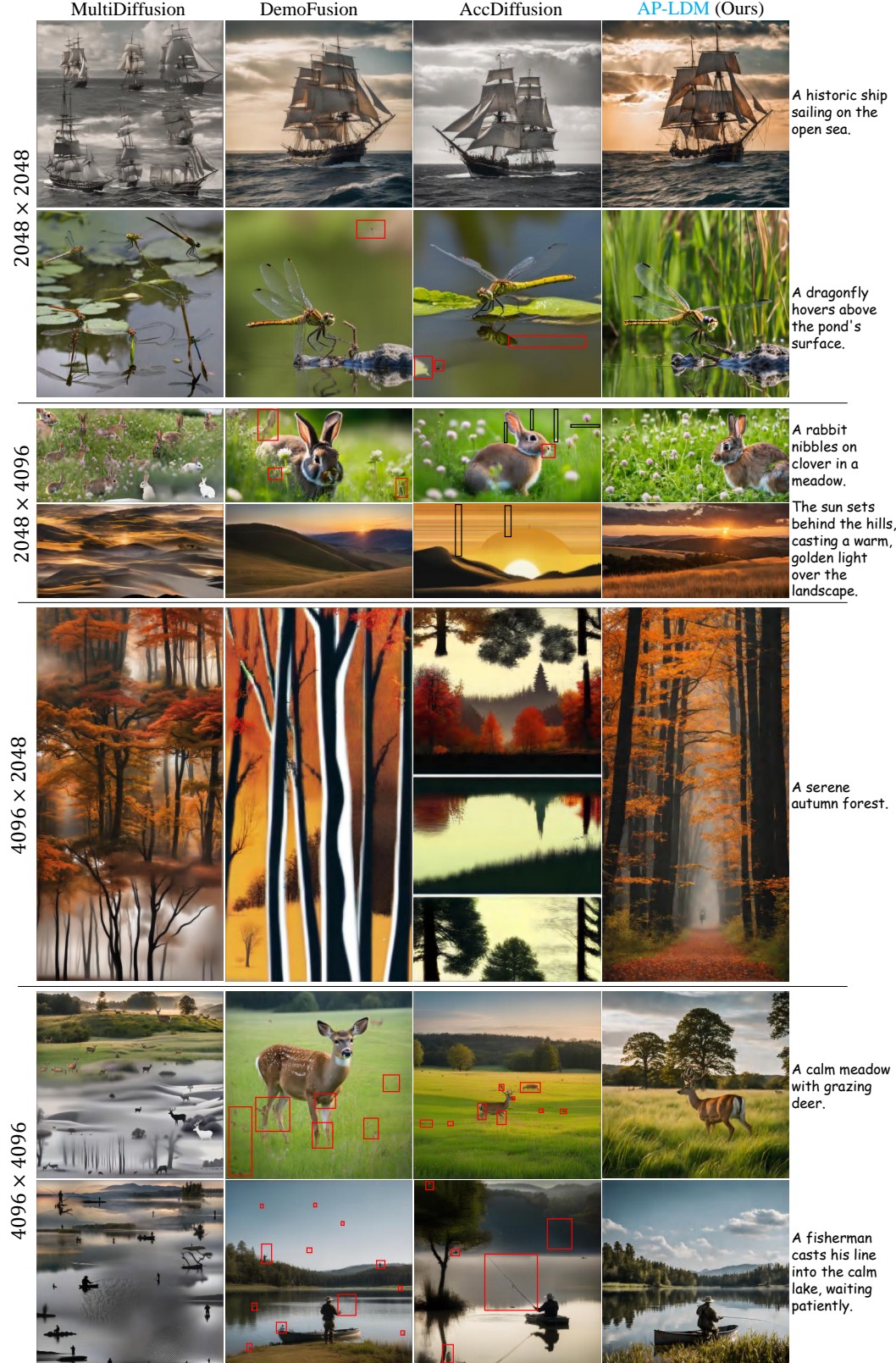

Figure 13: **Qualitative comparison with other baselines.**

### A.4 ANALYSIS OF THE PROGRESSIVE UPSAMPLING GENERATION PROCESS IN AP-LDM

To clearly illustrate the progressive upsampling process of AP-LDM, we set $\eta_2 = [0.2, 0.2, 0.2]$ to generate $4096 \times 4096$ images. As shown in Fig. 14, the images generated at different sub-stages of AP-LDM exhibit a high degree of consistency, with only minor differences in details. Since our task focuses on generating HR images rather than traditional image super-resolution, these differences in details are reasonable.

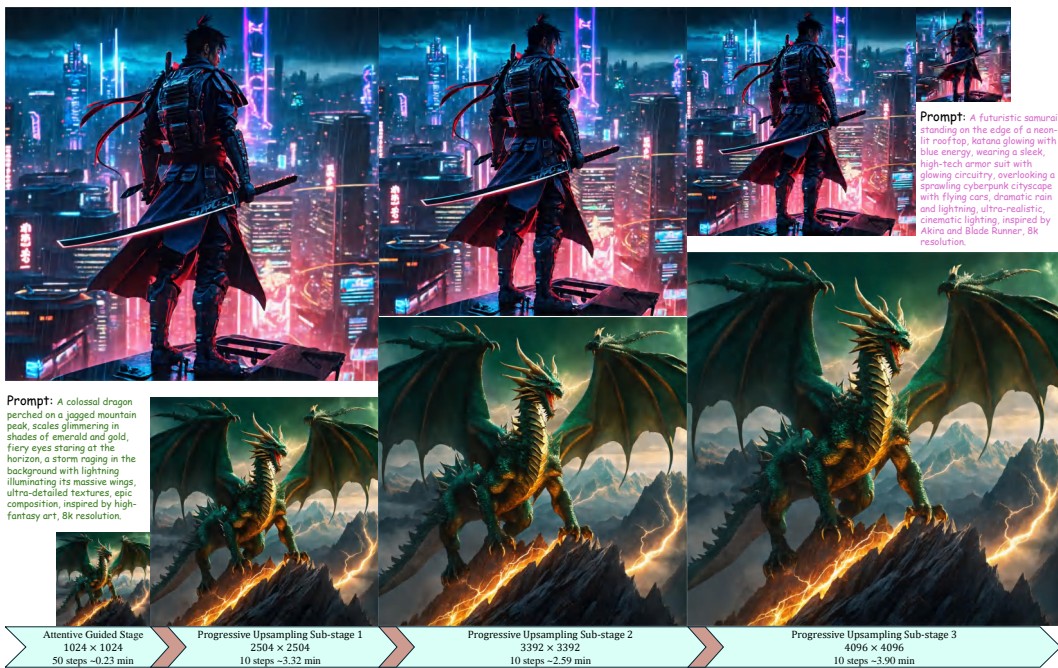

Figure 14: **Illustration of the progressive upsampling generation process.** The inference speed is evaluated on a single NVIDIA 3090 GPU.

Another noteworthy observation is that even though the progressive upsampling generation sub-stages involve only a small number of denoising steps (*e.g.*, 10 steps), the majority of the generation time is still consumed in these sub-stages. This is because the time required for denoising models to perform inference increases dramatically with the image size. For each denoising step, the time required for HR images is several times that for low-resolution images. Consequently, repeating a full denoising process at high resolution is extremely time-consuming (Du et al., 2024; Lin et al., 2024). Considering that HR and low-resolution images should share the same low-frequency structure, and that DMs naturally generate low-frequency structures first during denoising (Yu et al., 2023; Teng et al., 2023), AP-LDM effectively leverages the prior knowledge of low-frequency structures in low-resolution images. This significantly reduces the number of denoising steps needed at high resolution, thereby substantially accelerating the image generation process.

### A.5 HOW DOES PFSA WORK?

In this section, we further elaborate on the working mechanism of PFSA. Specifically, the functionality of PFSA can be described in two aspects: (**i**) clustering the related tokens in the latent representations; (**ii**) adjusting the amplitude of the high-frequency and low-frequency components in the latent representations.

#### A.5.1 PFSA CLUSTERS TOKENS OF LATENT REPRESENTATIONS

PFSA reorganizes tokens based on their similarities. Intuitively, this enables PFSA to perform token clustering, which enhances the structural consistency of latent representations. To demonstrate the clustering effect of PFSA, we calculated the deviation of the tokens' mean (DTM) of the latent representations $\tilde{z}_t$ and $z_t$. Concretely, assuming $z_t \in \mathbb{R}^{h \times w \times c}$, and $Z_t = \text{Flatten}(z_t) = [\mathbf{y}_{t1}, \dots, \mathbf{y}_{tN}] \in \mathbb{R}^{N \times c}$, where $N = h \times w$, we calculate DTM as:

$$\text{DTM} = [\text{mean}(\boldsymbol{y}_{ti}) - \text{mean}(\boldsymbol{Z}_t) \quad \text{for} \quad i = 1, \ldots, N] \tag{5}$$

To provide an intuitive illustration of the clustering effect of PFSA, we visualize the DTM based on token indices (*i.e.*, $i = 1, \ldots, N$) when $t$ is relatively large. As shown in columns (A) and (B) of Fig. 15, compared to the DTM of $\boldsymbol{z}_t$ (blue points), the DTM of $\tilde{\boldsymbol{z}}_t$ (red points) becomes more dispersed and exhibits distinct stripe patterns, indicating that PFSA indeed clusters the tokens of the latent representations. This clustering effect can be more directly demonstrated when $t$ is smaller. As shown in the heatmaps in columns (C) and (D) of Fig. 15, it is evident that PFSA clusters semantically related tokens.

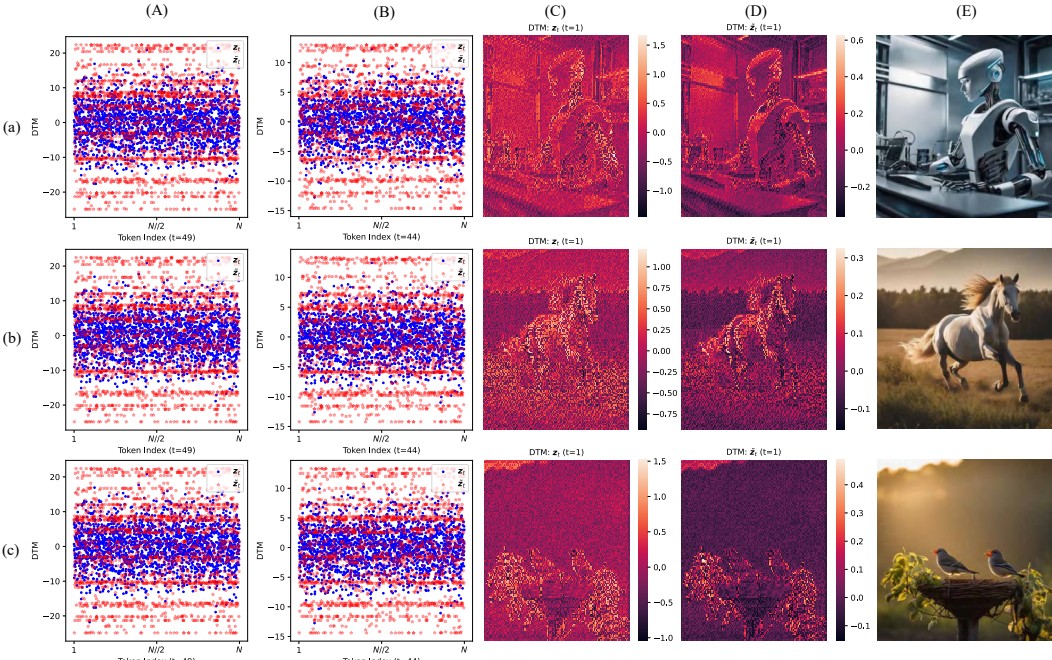

Figure 15: **The clustering effect of PFSA.** Columns (A), (B), (C), and (D) show the DTM of latent representations, while column (E) presents the corresponding generated RGB images.

### A.5.2 PFSA ADJUSTS THE AMPLITUDE OF HIGH- AND LOW-FREQUENCY COMPONENTS IN LATENT REPRESENTATIONS

The aim of this section is to explain: (**i**) why appropriately delaying attentive guidance can resolve structural deformation issues (as shown in Fig. 8), (**ii**) why attentive guidance enhances the details and colors of the image (as shown in Fig. 6, 7, and 12), and (**iii**) why applying attentive guidance in the later stages of denoising does not enhance the image details and colors (as shown in Fig. 9).

To explain the aforementioned three points, as shown in Fig. 16, we calculate the Fourier transforms of $\boldsymbol{z}_t$ (blue solid line) and $\tilde{\boldsymbol{z}}_t$ (red solid line), along with the mean of the standard deviations for all their channels (dashed line). It can be observed that PFSA significantly alters the relative amplitudes of the high- and low-frequency components in the latent representations during the initial denoising steps (from $t = 49$ to $t = 47$), particularly affecting the low-frequency components, which results in structural deformation. During the early and middle stages of denoising (from $t = 44$ to $t = 29$), PFSA increases the amplitudes of high-frequency components in the latent representations, which explains why attentive guidance leads to richer details and colors. In the later stages of denoising (from $t = 28$ to $t = 0$), PFSA slightly suppresses the high-frequency components of the latent representations while almost leaving the low-frequency components unchanged. This explains why applying attentive guidance in the later stages of denoising cannot enrich details and colors of the generated images.

Additionally, Fig. 16 shows that PFSA increases the standard deviation of $\tilde{\boldsymbol{z}}_t$ during the early and middle stages of denoising, while decreasing it in the later stages. The trend of the standard deviation

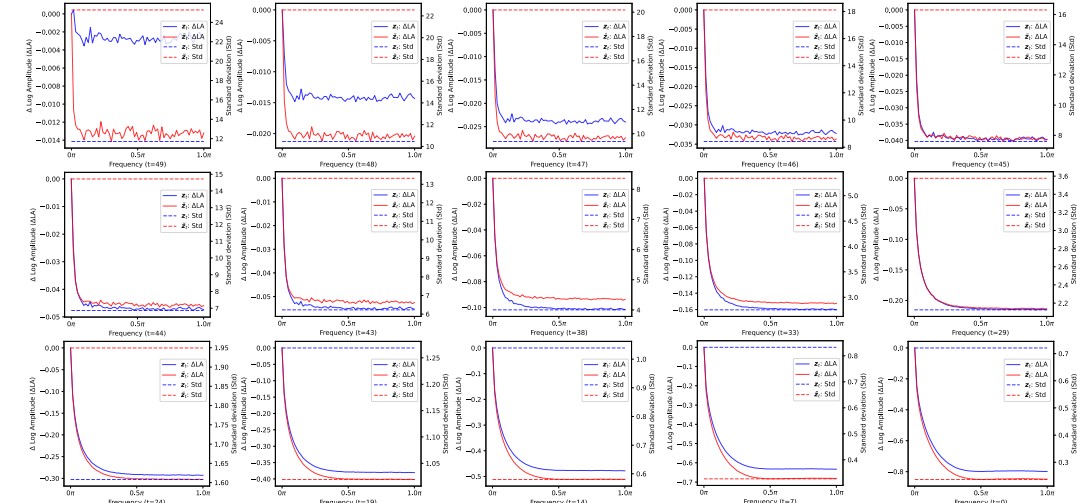

Figure 16: **The Fourier transform of the latent representation and the mean of the standard deviations across all channels.** $z_t$ is represented in blue, while $\tilde{z}_t$ is represented in red; the Fourier transforms are shown as solid lines, and the standard deviations are shown as dashed lines. The results are based on the generation process of 5k images.

changes is closely consistent with the variation in the amplitude of the high-frequency components. We conjecture that this is because the amount of information in the latent representations is positively correlated with the standard deviation, where a larger standard deviation corresponds to more image details and larger high-frequency components.

## A.6 COMPARISON WITH ADDITIONAL BASELINE MODELS

In this section, we compare AP-LDM with additional baseline models. Specifically, we include recently proposed HiDiffusion (Zhang et al., 2025) and a super-resolution model (SDXL+BSRGAN, *i.e.*, the outputs of SDXL are upsampled using BSRGAN (Zhang et al., 2021)). Since HiDiffusion experiments are conducted using professional-grade V100 GPUs without optimization for ultra-high-resolution images, it is not feasible to generate images with resolutions above $2048 \times 4096$ on consumer-grade GPUs such as the 3090. Due to device limits, we compare its performance only at the resolution of $2048 \times 2048$. The experimental setup remains the same as described in §4.

### A.6.1 QUANTITATIVE COMPARISON

**Comparison of generated image quality.** Table 8 presents the extended quantitative comparison results of generated image quality, which further demonstrate our effectiveness on HR image generation. We observe that models employing progressive upsampling generation (*e.g.*, AP-LDM, DemoFusion, and AccDiffusion) achieved relatively better results, showing the robustness of the progressive upsampling generation paradigm.

Table 8: **Quantitative comparison results**. The best results are marked in **bold**, and the second best results are marked by underline.

| Method | $2048 \times 2048$ | | | | | $2048 \times 4096$ | | | | | $4096 \times 2048$ | | | | | $4096 \times 4096$ | | | | |
|---|---|---|---|---|---|---|---|---|---|---|---|---|---|---|---|---|---|---|---|---|
| | FID | IS | FID$_c$ | IS$_c$ | CLIP | FID | IS | FID$_c$ | IS$_c$ | CLIP | FID | IS | FID$_c$ | IS$_c$ | CLIP | FID | IS | FID$_c$ | IS$_c$ | CLIP |
| SDXL | 99.9 | 14.2 | 80.0 | 16.9 | 25.0 | 149.9 | 9.5 | 106.3 | 12.0 | 24.4 | 173.1 | 9.1 | 108.5 | 11.5 | 23.9 | 191.4 | 8.3 | 114.1 | 12.4 | 22.9 |
| MultiDiff. | 99.8 | 14.5 | 67.9 | 17.1 | 24.6 | 125.8 | 9.6 | 71.9 | 15.7 | 24.6 | 149.0 | 9.0 | 70.5 | 14.4 | 24.4 | 168.4 | 6.5 | 76.6 | 14.4 | 23.1 |
| ScaleCrafter | 98.2 | 14.2 | 89.7 | 13.3 | 25.4 | 161.9 | 10.0 | 154.3 | 7.5 | 23.3 | 175.1 | 9.7 | 167.3 | 8.0 | 21.6 | 164.5 | 9.4 | 170.1 | 7.3 | 22.3 |
| UG | 82.2 | 17.6 | 65.8 | 14.6 | 25.5 | 155.7 | 8.2 | 165.0 | 6.6 | 21.7 | 185.3 | 6.8 | 175.7 | 6.2 | 20.5 | 187.3 | 7.0 | 197.6 | 6.3 | 21.8 |
| DemoFusion | 72.3 | **21.6** | 53.5 | **19.1** | 25.2 | 96.3 | 17.7 | 62.3 | 15.0 | 25.0 | 99.6 | 16.4 | 61.9 | 14.7 | 24.4 | 101.4 | 20.7 | 63.5 | 13.5 | 24.7 |
| AccDiff. | 71.6 | 21.0 | 52.7 | 17.0 | 25.1 | 95.5 | 16.4 | 62.9 | 11.1 | 24.5 | 102.2 | 15.2 | 65.4 | 11.5 | 24.2 | 103.2 | 20.1 | 65.9 | 13.3 | 24.6 |
| SDXL+BSR. | 66.2 | 21.1 | 47.5 | 16.6 | **25.7** | **80.7** | 19.8 | 50.2 | 12.3 | 25.1 | **92.7** | 17.6 | 57.9 | 12.1 | 24.9 | 90.0 | 20.9 | 56.0 | 13.8 | **25.2** |
| HiDiff. | 81.0 | 16.8 | 64.1 | 14.2 | 24.9 | - | - | - | - | - | - | - | - | - | - | - | - | - | - | - |
| AP-LDM | **66.0** | 21.0 | **47.4** | 17.5 | 25.1 | 89.0 | **20.3** | 56.0 | **19.0** | 25.0 | 93.2 | **19.5** | 56.9 | **16.5** | 24.9 | 90.6 | **21.1** | 59.0 | **14.8** | 24.6 |

In contrast, HiDiffusion fell short compared to methods using progressive upsampling. We speculate that its suboptimal performance is due to two factors: (**i**) the forced resizing of deep feature maps during the generation process, which causes significant distribution shifts; and (**ii**) the use of MSW-MSA (a sparse attention mechanism similar to SwinTransformer (Liu et al., 2021)), which forcibly

alters the attention's receptive field and sequence length, leading to severe shifts in the entropy of attention weights (Jin et al., 2024). The aforementioned two issues prevent HiDiffusion from fully addressing the problem of repeated object structures and result in severe artifacts and deformations in the generated images (as shown in Fig. 17).

The super-resolution model (SDXL + BSRGAN) demonstrated strong performance in quantitative experiments, a phenomenon also observed in the DemoFusion's experiments. This is because super-resolution models can at least preserve the low-frequency structures of images without significant errors. However, as discussed in DemoFusion (Du et al., 2024) and AccDiffusion (Lin et al., 2024), super-resolution models fail to add finer details to high-resolution images (as shown in Fig. 18).

**Comparison of resource consumption.** We also compare the inference time and GPU memory usage required by the models. Specifically, we test the minimum GPU memory requirements during model inference based on the model's open-source code. Table 9 shows the resource consumption of different models when generating images at various resolutions. SDXL+BSRGAN, unlike DMs, does not require iterative inference, allowing it to achieve the fastest generation speed. However, the super-resolution model fails to generate the level of detail expected in high-resolution images, which has limited its widespread adoption.

Table 9: **Model resource consumption**. The best results are marked in **bold**, and the second best results are marked by underline. Time unit: minute. Storage unit: GB.

| Method | $2048 \times 2048$ | | $2048 \times 4096$ | | $4096 \times 4096$ | |
|---|---|---|---|---|---|---|
| | time cost | storage cost | time cost | storage cost | time cost | storage cost |
| SDXL | 1.0 | 15.9 | 3.0 | 16.1 | 8.0 | **16.6** |
| MultiDiff. | 3.0 | 22.0 | 6.0 | 16.8 | 15.0 | 16.8 |
| ScaleCrafter | 1.0 | 17.4 | 6.0 | 17.6 | 19.0 | 19.1 |
| UG | 1.8 | 23.9 | 4.0 | 16.5 | 11.1 | 18.0 |
| DemoFusion | 3.0 | 15.2 | 11.0 | 18.4 | 25.0 | 16.8 |
| AccDiff. | 3.0 | 22.1 | 12.7 | 23.0 | 26.0 | 22.1 |
| SDXL+BSR. | 1.0 | **14.6** | **1.0** | **11.1** | **1.0** | 21.1 |
| HiDiff. | 0.8 | 23.9 | - | - | - | - |
| AP-LDM | **0.6** | 16.0 | 2.0 | 21.1 | 5.7 | 23.8 |

It is worth noting that for high-resolution image generation tasks, the memory bottleneck lies in the encoding and decoding of the VAE rather than interpolating the image in pixel space. To address the challenges of encoding and decoding high-resolution images, researchers typically employ tiled encoders and tiled decoders. In this work, we also utilize a tiled-encoder and decoder when generating ultra-high-resolution images, allowing us to generate images with resolutions up to $4096 \times 7280$ or higher on a 24GB VRAM NVIDIA 3090 GPU (as shown in Fig. 1).

### A.6.2 QUALITATIVE COMPARISON

**Qualitative Comparison with HiDiffusion.** We conduct extensive qualitative comparison experiments between AP-LDM and HiDiffusion, with the results shown in Fig. 17. From the figure, it can be observed that AP-LDM consistently generates high-quality, high-resolution images. Although capable of generating some good results, HiDiffusion suffers from significant distribution shifts in the UNet features due to forced feature scaling and the use of window attention, which alters the sequence length during attention computation. This often causes the generated images to collapse, as illustrated in Fig. 17 (a)–(e). Even when HiDiffusion avoids image collapse, it frequently produces noticeable artifacts and distortions, as shown in Fig. 17 (f)–(h). In Fig. 17 (i) and (j), HiDiffusion still exhibits severe structural repetition in the generated outputs, indicating that merely resizing the deep features of the UNet is insufficient to completely eliminate low-frequency structural errors.

**Qualitative Comparison with SDXL+BSRGAN.** We conducted extensive qualitative comparisons between AP-LDM and SDXL+BSRGAN. Specifically, we compared their performance at resolutions of $2048 \times 2048$ (Fig. 18 (a)-(d)) and $4096 \times 4096$ (Fig. 18 (e)-(h)). As we can see, compared to AP-LDM, SDXL+BSRGAN, while maintaining decent image structure, fails to generate the level of detail expected from HR images. The absence of these details sometimes leads to the model's inability to simulate realistic scenes. For example, in Fig. 18 (c), SDXL+BSRGAN fails to generate realistic shadows. At higher resolutions (*e.g.*, $4096 \times 4096$), SDXL+BSRGAN may introduce artifacts, as shown in Fig. 18 (e) and (g).

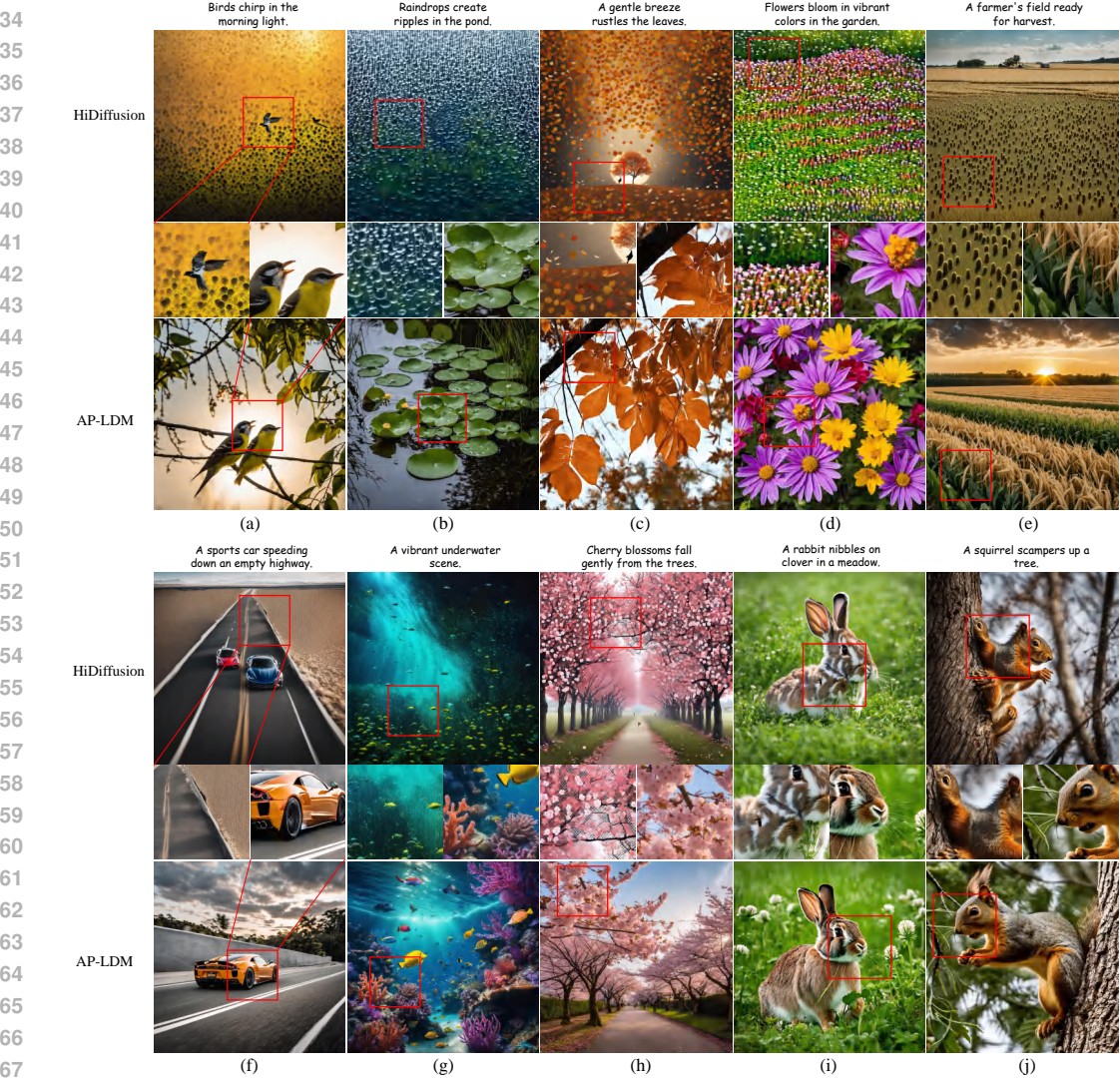

Figure 17: **Quantitative comparison with HiDiffusion**, where all images have a resolution of $2048 \times 2048$. The prompts for the generated images are provided above the figures.

### A.7 ATTENTIVE GUIDANCE ALSO WORKS IN OTHER GENERATION FRAMEWORKS

In this section, we apply attentive guidance to other generative frameworks to demonstrate its generalization capability. Specifically, we apply attentive guidance to the generative frameworks of HiDiffusion and DemoFusion, and conduct both quantitative and qualitative ablation studies.

#### A.7.1 QUANTITATIVE ABLATION IN OTHER GENERATIVE FRAMEWORKS

In this section, considering the long inference time of DemoFusion, we perform quantitative ablation studies on attentive guidance using the HiDiffusion generation frameworks at a resolution of $2048 \times 2048$. All experimental settings are consistent with those in §4.

Table 10: **Quantitative ablation of attentive guidance using HiDiffusion frameworks**. The best results are marked in bold. AG: attentive guidance.

| Method | FID | IS | FID$_c$ | IS$_c$ | CLIP |
|---|---|---|---|---|---|
| HiDiffusion | 81.0 | 16.8 | 64.1 | 14.2 | **24.9** |
| HiDiff.+AG | **79.4** | **17.0** | **62.4** | **14.6** | 24.9 |

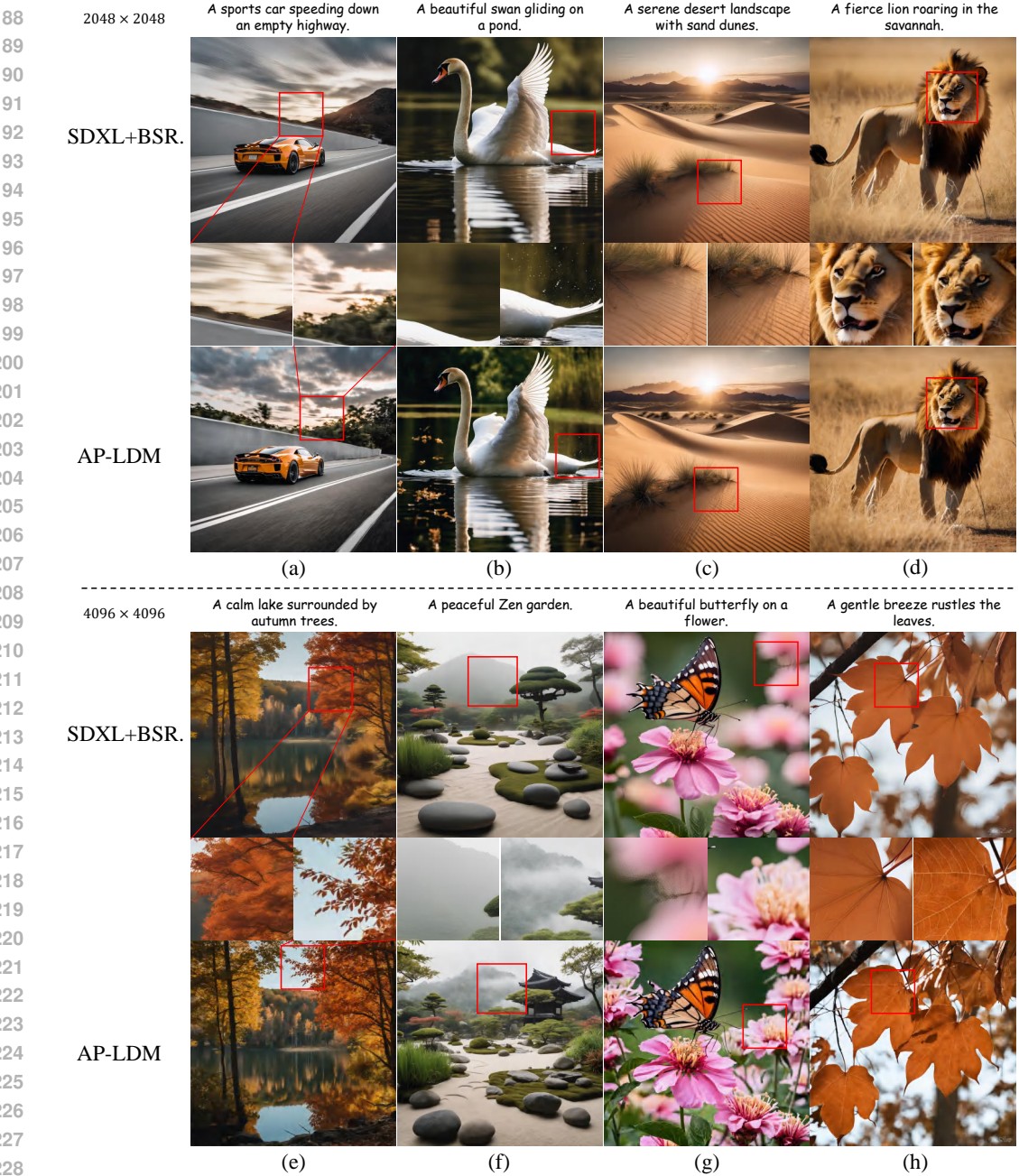

Figure 18: **Qualitative comparison with SDXL+BSRGAN**. Figures (a)-(d) have a resolution of $2048 \times 2048$, while Figures (e)-(h) have a resolution of $4096 \times 4096$. The prompts for the generated images are provided above the figures.

Table 10 presents the quantitative ablation results using the HiDiffusion framework. It is evident that incorporating attentive guidance improves HiDiffusion across all metrics. This is further corroborated by the qualitative analysis in Fig. 19, which demonstrates that attentive guidance alleviates some of the structural collapses observed in HiDiffusion.

### A.7.2 QUALITATIVE ABLATION STUDIES IN OTHER GENERATIVE FRAMEWORKS

**HiDiffusion+attentive guidance.** HiDiffusion enforces scaling of the UNet feature maps during image generation, which often leads to structural collapse and deformations in the generated images (as shown in Fig. 17). Fig. 19 (a)-(f) demonstrate that using attentive guidance effectively mitigates

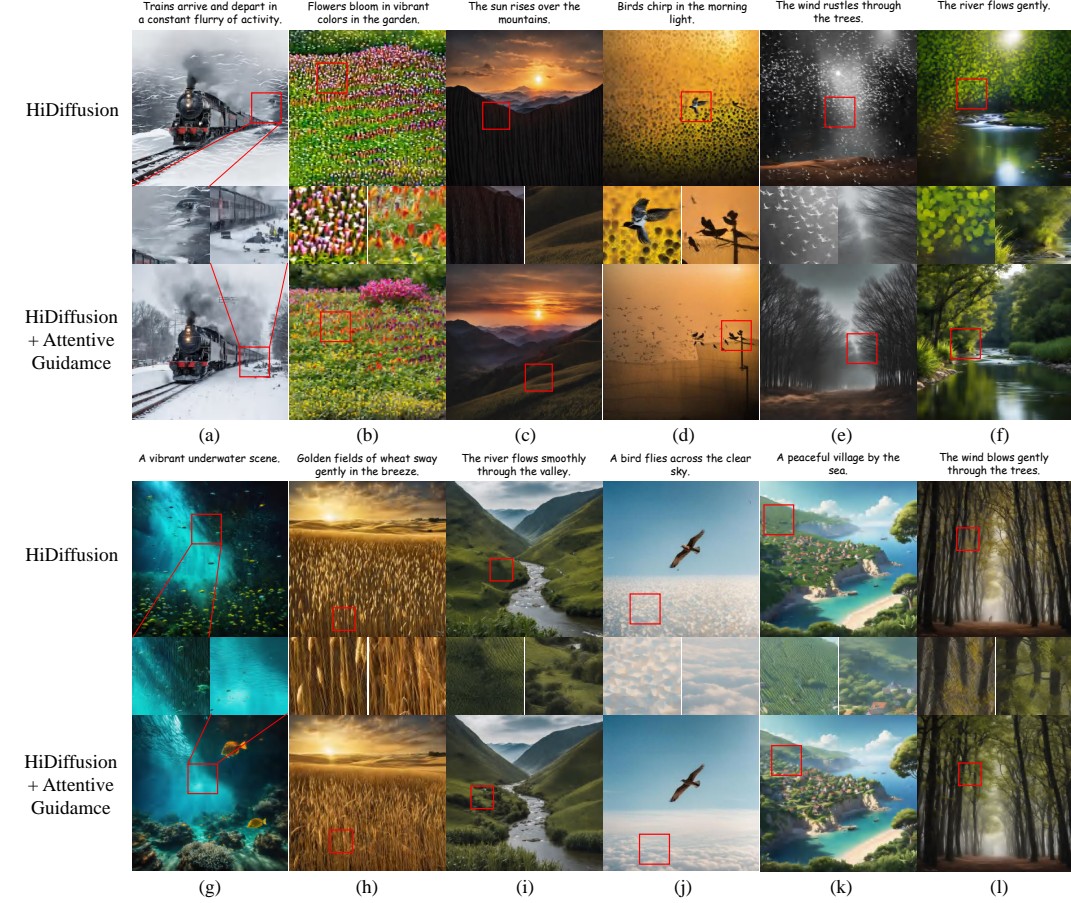

Figure 19: **Qualitative ablation of attentive guidance in the HiDiffusion Framework**. All images have a resolution of $2048 \times 2048$. Figures (a)-(f) demonstrate that attentive guidance can mitigate the issue of structural collapse in generated images, while Figures (g)-(l) show that attentive guidance resolves structural deformation issues and enhances image details.

the issue of structural collapse in synthesized images. Fig. 19 (g)-(l) further show that attentive guidance can also address the structural deformation inherent to HiDiffusion, enhance image details, and improve overall image quality.

**DemoFusion+attentive guidance.** In the analysis presented in §4.3 and §A.3, we observed that DemoFusion tends to produce repetitive structures (as shown in Fig. 5 and 13), a phenomenon also noted in other studies (Lin et al., 2024). We incorporate attentive guidance into the generative framework of DemoFusion. As shown in Fig. 20 (a)-(e), attentive guidance effectively mitigates the issue of repetitive structures in DemoFusion. Fig. 20 (f)-(j) further illustrate role of attentive guidance in enriching image details and enhancing overall image quality.

## A.8    COMPARATIVE AND ABLATION ANALYSIS BASED ON STABLEDIFFUSION 2.1

To validate the generalization capability of AP-LDM, we conducted extensive quantitative and qualitative analyses using StableDiffusion 2.1 (SD2.1) as the pretrained base model.

### A.8.1    COMPARISON EXPERIMENTS

**Quantitative comparison.** Since the code for using SD2.1 as the pretrained model in AccDiffusion and DemoFusion is not publicly available, we compare AP-LDM with ScaleCrafter in this section. We compared the model performance at four resolutions: $1536 \times 1536$, $1024 \times 2048$, $2048 \times 1024$, and $2048 \times 2048$. Considering that SD2.1's generation capabilities are weaker than

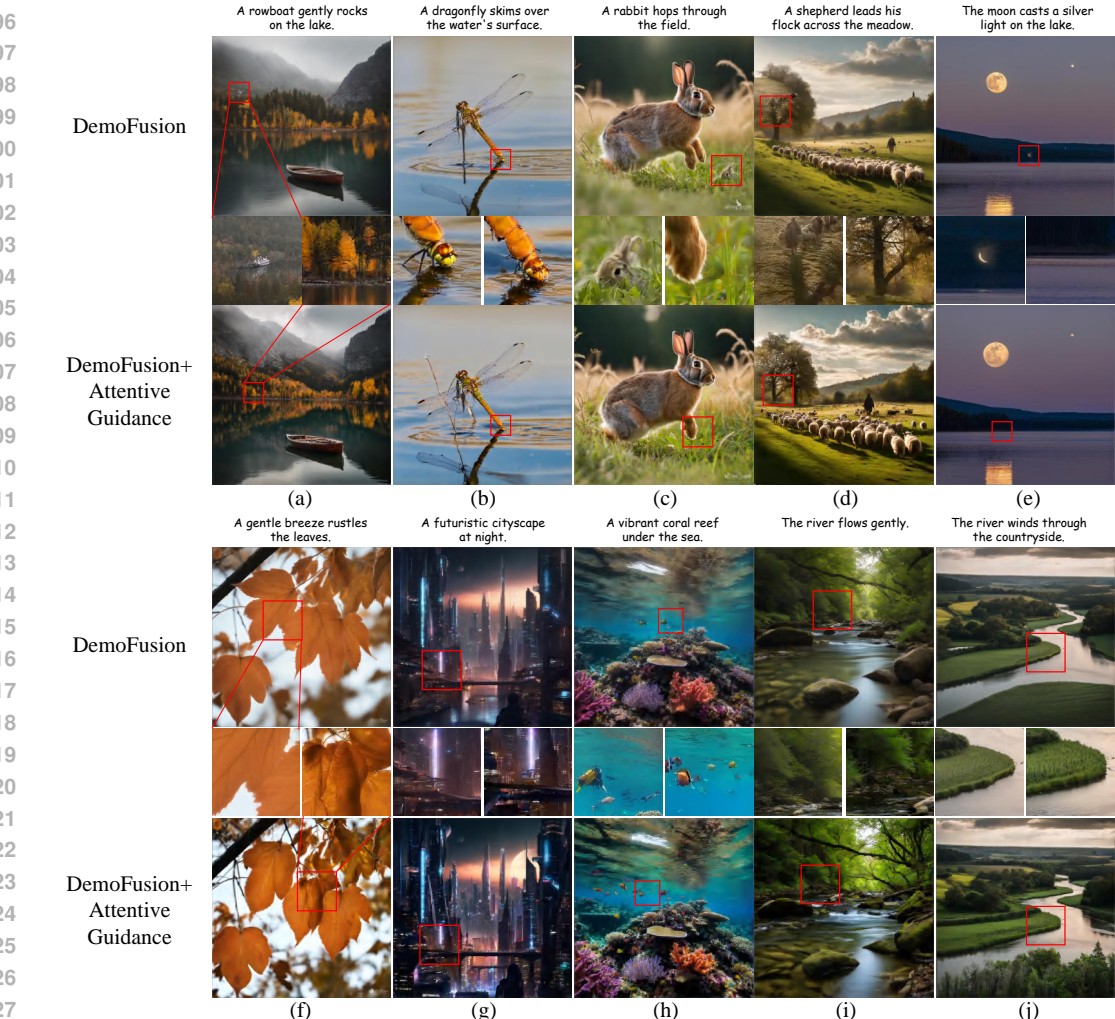

Figure 20: **Qualitative ablation of attentive guidance in the DemoFusion Framework**. All images have a resolution of $2048 \times 2048$. Figures (a)-(e) demonstrate that attentive guidance effectively mitigates the issue of repetitive structures in images, while Figures (f)-(j) showcase attentive guidance's ability to enrich image details.

SDXL, we set $\eta_2 = [0.2, 0.2, 0.3]$ for the experiments in this section, while keeping other settings consistent with §4.

Table 11: **Quantitative comparison results based on SD2.1**. The best results are marked in **bold**.

| Method | $1536 \times 1536$ | | | | | $1024 \times 2048$ | | | | | $2048 \times 1024$ | | | | | $2048 \times 2048$ | | | | |
|---|---|---|---|---|---|---|---|---|---|---|---|---|---|---|---|---|---|---|---|---|
| | FID | IS | FID$_c$ | IS$_c$ | CLIP | FID | IS | FID$_c$ | IS$_c$ | CLIP | FID | IS | FID$_c$ | IS$_c$ | CLIP | FID | IS | FID$_c$ | IS$_c$ | CLIP |
| SD2.1 | 95.4 | 17.8 | 83.4 | 15.8 | 25.0 | 85.8 | 15.9 | 76.1 | 16.3 | **25.2** | 101.8 | 15.8 | 79.8 | 16.8 | 24.6 | 121.7 | 14.4 | 92.7 | 14.4 | 24.5 |
| ScaleCrafter | 140.4 | 10.6 | 136.4 | 9.7 | 21.9 | 150.0 | 10.1 | 139.3 | 10.1 | 21.7 | 149.8 | 10.4 | 135.6 | 11.5 | 21.8 | 144.2 | 10.4 | 135.2 | 10.3 | 23.4 |
| AP-LDM | **60.3** | **21.0** | **50.6** | **18.3** | **25.4** | **61.1** | **19.9** | **54.1** | **18.4** | 25.0 | **63.7** | **19.2** | **50.4** | **18.2** | **24.7** | **60.5** | **21.5** | **48.8** | **17.2** | **25.3** |

Table 11 presents the results of the quantitative comparison, demonstrating that AP-LDM maintains strong performance when using SD2.1 as the pre-trained model. ScaleCrafter, on the other hand, performs suboptimally due to its tendency to produce structural collapse in generated images, a phenomenon more evident in the qualitative analysis.

**Qualitative comparison.** Fig. 21 presents the results of the qualitative comparison. It can be observed that when generating high-resolution images, SD2.1 also encounters issues with repetitive object structures. ScaleCrafter frequently exhibits structural collapse in generated images during denoising with SD2.1, leading to its suboptimal performance. In contrast, AP-LDM consistently produces high-quality results across all resolutions, demonstrating the generalizability of the AP-LDM generation framework.

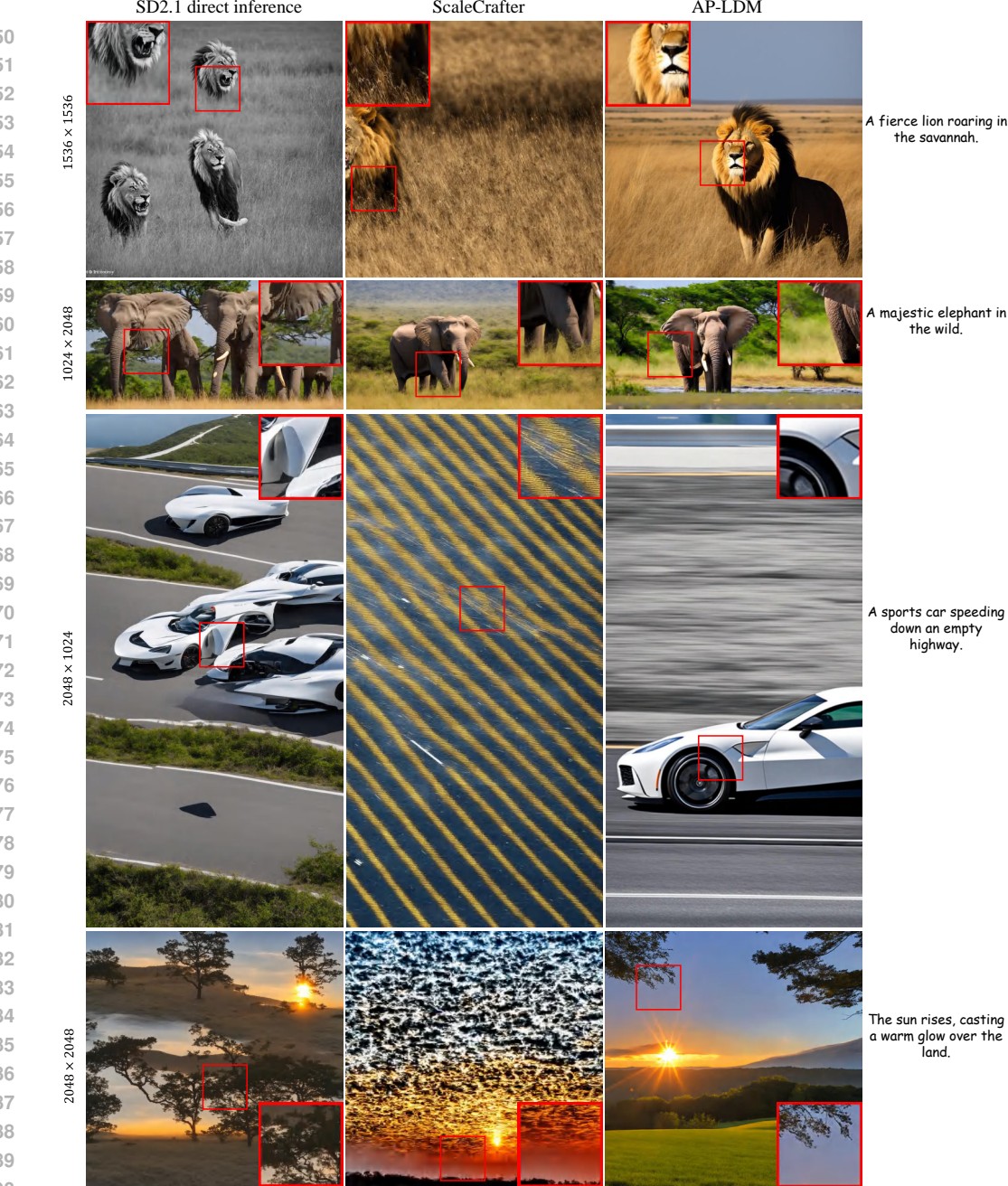

Figure 21: **Qualitative comparison using SD2.1 as the pretrained model**.

### A.8.2 ABLATION STUDY ON ATTENTIVE GUIDANCE

**Quantitative ablation.** Table 12 shows the results of the quantitative ablation on attentive guidance using SD2.1 as the pretrained model. It can be observed that attentive guidance leads to improvements in metrics. These improvements are more evident in the qualitative ablation analysis.

Table 12: **Quantitative ablation results based on SD2.1**. The best results are marked in **bold**.

| Method | 1536 × 1536 | | | | | 1024 × 2048 | | | | | 2048 × 1024 | | | | | 2048 × 2048 | | | | |
|---|---|---|---|---|---|---|---|---|---|---|---|---|---|---|---|---|---|---|---|---|
| | FID | IS | $FID_c$ | $IS_c$ | CLIP | FID | IS | $FID_c$ | $IS_c$ | CLIP | FID | IS | $FID_c$ | $IS_c$ | CLIP | FID | IS | $FID_c$ | $IS_c$ | CLIP |
| w/o AG | 61.2 | 20.9 | **50.2** | **18.9** | 25.2 | 61.5 | 19.6 | **54.0** | **19.5** | 24.9 | 64.6 | **19.6** | 49.2 | 17.0 | 24.6 | 61.1 | 21.2 | **46.5** | **18.2** | 25.2 |
| w/ AG | **60.3** | **21.0** | 50.6 | 18.3 | **25.4** | **61.1** | **19.9** | 54.1 | 18.4 | **25.0** | **63.7** | 19.2 | 50.4 | **18.2** | **24.7** | **60.5** | **21.5** | 48.8 | 17.2 | **25.3** |

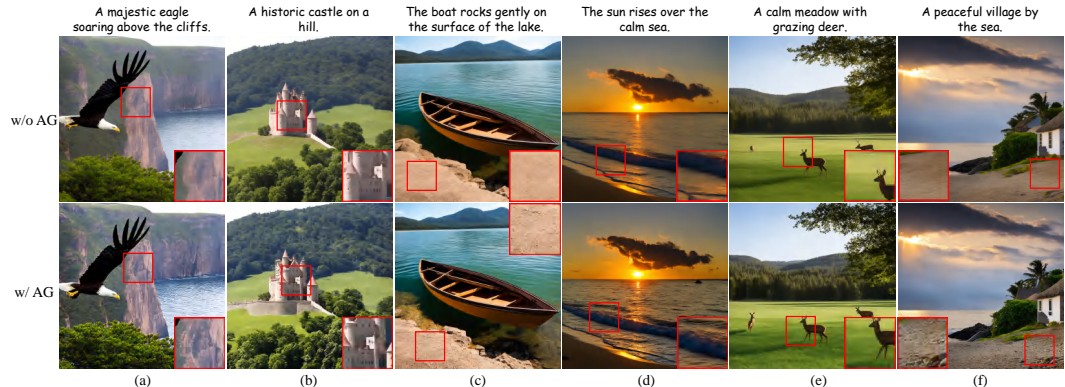

Figure 22: **Ablation study of attentive guidance using SD2.1 as the pre-trained model**. Resolution: $2048 \times 2048$.

**Qualitative ablation.**   Fig. 22 presents the ablation analysis of attentive guidance based on SD2.1. From the figure, it can be observed that attentive guidance also enhances detail richness and color vibrancy when using SD2.1, further demonstrating its generalization capability.

