# OpenReview forum: "AP-LDM: Attentive and Progressive Latent Diffusion Model for Training-Free High-Resolution Image Generation"
_ICLR.cc/2025/Conference — Submitted to ICLR 2025_

### Official Review · Reviewer_9bp6 · 2024-10-31

**Soundness:** 3
**Presentation:** 2
**Contribution:** 3
**Rating:** 6
**Confidence:** 4

**Summary:**

The authors propose a training-free framework, Attentive and Progressive LDM (AP-LDM), for high-resolution image generation. Specifically, they introduce attentive training-resolution (TR) denoising and progressive high-resolution denoising. The model effectively enhances image detail by using the parameter-free self-attention mechanism (PFSA). The proposed approach achieves superior quality and generation speed compared to existing methods.

**Strengths:**

1. The authors provide a reasonable analysis of the limitations of existing training-free frameworks and address several shortcomings.
2. The proposed PFSA and progressive high-resolution denoising are both simple and effective, and the motivations provided are logical.
3. Extensive quantitative and qualitative comparisons are presented to demonstrate the effectiveness of the proposed method.
4. Compared to other training-free frameworks, the proposed method shows significant advantages in both image quality and generation speed.

**Weaknesses:**

1. The authors conduct experiments on only one model; it is recommended to try more baselines. Given that SDXL is a model with a large number of parameters (over 10B), it would help to further demonstrate the generalizability of the proposed method by testing on lighter models, such as Stable-Diffusion-v2.1 (1B).
2. There is no comparison of GPU memory usage. Considering that diffusing on the upsampled image requires more memory, this comparison is necessary. The authors should provide memory usage data for different methods across resolutions.
3. The authors do not perform ablation studies on the position of the AG module, such as at the input or output stage.
4. For visual comparison, it is suggested to enlarge the red boxes and provide a close-up for easier readability, as in the examples in Figures 5 and 8.

**Questions:**

1. How does the method perform on other baselines? Is it sufficiently generalizable?
2. What is the specific memory usage? Does it have any advantages over other methods?
3. Other questions are noted in the Weaknesses section.

---

> ### Author Response · Authors · 2024-11-28
> **Response to Reviewer 9bp6**
>
> # About Supplementary Materials
> Our full paper file exceeds 100MB. Due to upload size limitations, we have submitted a compressed version of the paper. To ensure clear visualization of the images, **we have uploaded pages 18–27 of the appendix as supplementary materials**. Please refer to these pages for further details!
>
> # Response to Weaknesses
> 1. Following your suggestion, we conduct further comparisons and ablation experiments using the SD2.1 pre-trained model, as presented in Appendix A.8 of the revised paper. **The experimental results demonstrate the generalization capability of AP-LDM**. In the future, we will use SD2.1 to reproduce more models and conduct a more comprehensive comparison.
>
> 2. In Table 9 of Appendix A.6.1 in the revised paper, we provide memory usage details for different models when generating images at various resolutions. Some models exhibit lower memory usage for high-resolution image generation compared to lower resolutions due to the use of a tiled decoder for high-resolution images. This indicates that **the memory bottleneck during high-resolution image generation lies not in the denoising process but in the VAE decoding process**.
> **In AP-LDM, the pixel space upsampling process is also not the memory bottleneck; instead, the bottleneck is in the VAE encoding process**. To address this, **we employed a tiled encoder for pixel encoding**. As a result, while AP-LDM does not offer additional memory savings compared to other methods, **it does enable the synthesis of ultra-high-resolution images—such as 4096x7280 or even higher—on consumer-grade GPUs like the 3090**, as shown in Figure 1.
>
> 3. The goal of attentive guidance is to enhance the structural consistency of the latent representations, directly operating on the representations obtained after each denoising step. **In diffusion models, the latent representation at step t serves as the output of step t+1 and simultaneously as the input to step t-1**. In other words, **attentive guidance works on both the model input and the model output**, which is why we did not perform ablation studies specifically for it.
>
> 4. Following your suggestion, we have enlarged all the red bounding boxes in the paper. Due to space constraints in the main text, we have included close-up views in the appendix. You can find the comparisons with detailed close-ups in Figures 17-22 of the appendix. Due to time constraints, we will provide more close-up comparison results with other models in the next version.

---

> ### Author Response · Authors · 2024-11-29
> **Gentle Reminder**
>
> Dear Reviewer 9bp6,
>
> Thank you once again for your comprehensive and thoughtful feedback on our submission. As the discussion period nears its end, we are eager to know if our additional results and clarifications have adequately addressed your questions.
> We would sincerely appreciate any further perspectives or discussions you might have at this stage.
> Thank you for your time and engagement!
>
> Best regards,
>
> Authors

---

> > ### Comment · Reviewer_9bp6 · 2024-12-02
> >
> > Thank you for the response. The reply resolves my concerns regarding the method's generalization and memory usage. Therefore, I will maintain my current score.

---

> > > ### Author Response · Authors · 2024-12-02
> > >
> > > Dear reviewer 9bp6,
> > >
> > > Thank you for your thoughtful response and positive feedback. We truly appreciate the time and effort you’ve dedicated to reviewing our work and are glad that our clarifications and additional experiments have addressed your concerns. Your insights are valuable in improving our work.
> > >
> > > Sincerely,
> > >
> > > Authors

---

### Official Review · Reviewer_UP1g · 2024-10-31

**Soundness:** 3
**Presentation:** 3
**Contribution:** 3
**Rating:** 5
**Confidence:** 4

**Summary:**

This paper introduces AP-LDM, a simple but effective training-free framework for enhancing the quality and speed of HR image generation. AP-LDM divides the denoising process of latent diffusion models into two stages: (1) attentive training-resolution denoising using a parameter-free self-attention mechanism to improve structural consistency, and (2) progressive high-resolution denoising through pixel-space upsampling to reduce artifacts. Experimental results demonstrate that AP-LDM outperforms state-of-the-art methods in both image quality and inference speed, achieving up to a X5 speedup.

**Strengths:**

From the experimental results, attentive attention is simple yet effective, as it enhances the details during generation. This simple mechanism also lays the foundation for the fast generation of 4K resolution images in the future.

**Weaknesses:**

Although attentive attention is simple and effective, its internal working mechanism is not clearly presented. I have the following questions regarding how it functions:

[1] The authors propose attentive attention and pixel-domain progressive sampling for high-resolution image generation. While the latter clearly cannot handle multi-object scenarios, as shown by the MultiDiffusion results in Fig. 2(b), it remains unclear why attentive attention can address this issue. For instance, demofusion avoids this problem by introducing additional mechanisms like Skip Residual and Dilated Sampling. Clarification is needed on how attentive attention resolves multi-object issues.

[2] Attentive attention seems to have broader applications beyond high-resolution generation, as it is a training-free mechanism. This suggests it could potentially be used to enhance the quality and concept generation in regular image synthesis tasks. Further exploration of this would be valuable.

**Questions:**

[1] The paper lacks comparisons with recent works on efficient high-resolution generation, such as HiDiffusion.

[2] Lacking comparison with the baseline, SDXL + BSRGAN.

I will consider raising the score once all concerns are addressed.

---

> ### Author Response · Authors · 2024-11-28
> **Response to Reviewer UP1g**
>
> # About Supplementary Materials
> Our full paper file exceeds 100MB. Due to upload size limitations, we have submitted a compressed version of the paper. To ensure clear visualization of the images, **we have uploaded pages 18–27 of the appendix as supplementary materials**. Please refer to these pages for further details!
>
> # Response to weaknesses
> 1. In fact, structural and object repetition issues are **caused by the shift in the entropy of the attention maps in the Transformer layers during the early denoising stages, where low-frequency structures are generated first[1, 2, 3]**. Previous works utilized the latent representations of low-resolution images as conditions to guide the denoising of high-resolution images, or attempted to correct the entropy to alleviate this issue. We discard these previous paradigms, and **propose a novel PFSA serving to enhance the quality of low-resolution initialization images**. Specifically, PFSA improves the quality of low-resolution images by clustering related tokens in the latent representation and adjusting the relative strength of the high-frequency and low-frequency components. Then, AP-LDM uses pixel-space interpolation to upsample the resulting low-resolution image, adds noise, and performs denoising to obtain a high-quality high-resolution image. In this process, **AP-LDM avoids directly generating low-frequency structures in high-resolution images, thereby addressing the issue of object and structure repetition**. For a detailed explanation of the PFSA mechanism, please refer to Appendix A.5 in the revised paper.
>
> 2. **To lay the groundwork for future extension studies, we demonstrate the generalizability of attentive guidance in other generative frameworks** in Appendix A.7 of the revised paper.
> Specifically, we apply attentive guidance to HiDiffusion and DemoFusion for ablation studies to verify its generalizability. Extensive quantitative and qualitative results demonstrate that **attentive guidance significantly improves the performance of both HiDiffusion and DemoFusion**. Notably, attentive guidance not only enhances image details **but also effectively mitigates the structural collapse issues in HiDiffusion and the repetitive structure problems in DemoFusion**. This further demonstrates the effectiveness and generalizability of attentive guidance.
>
> # Response to Q1 and Q2
> Thank you for your valuable suggestions for enhancing our effectiveness. In Appendix A.6 of the revised paper, we provide extensive qualitative and quantitative comparisons of AP-LDM with HiDiffusion and the super-resolution method using BSRGAN.
> For the HiDiffusion with high hardware demanding, we can only generate images at 2048x2048 resolutions in our RTX 3090 GPUs instead of its required V100 GPUs.
> As shown in Table 8, **the quantitative results indicate that AP-LDM outperforms HiDiffusion, while SDXL+BSRGAN achieves quantitative results similar to those of AP-LDM**.
> In the qualitative analysis (Figure 17), we observed that **HiDiffusion's forced resizing of UNet feature maps often leads to structural breakdowns in the generated images, with frequent occurrences of structural and texture distortions**. We speculate that this is the main reason for HiDiffusion's suboptimal performance in the quantitative experiments.
> For the qualitative analysis of SDXL+BSRGAN (Figure 18), **although using BSRGAN for super-resolution does not provide sufficient image details, it consistently preserves the structural integrity of the images. We speculate that this is the reason for SDXL+BSRGAN's good performance in the quantitative experiments**. The authors of DemoFusion also compared their method with SDXL+BSRGAN, and the patterns we observed were consistent with their findings.
> Due to equipment limitations, we will later supplement the comparison results with HiDiffusion at other resolutions.
>
> # Reference
> [1] FreeDoM: Training-Free Energy-Guided Conditional Diffusion Model. ICCV 2023.\
> [2] Relay Diffusion: Unifying diffusion process across resolutions for image synthesis. ICLR 2024.\
> [3] Training-free Diffusion Model Adaptation for Variable-Sized Text-to-Image Synthesis. NeurIPS 2023.

---

> > ### Author Response · Authors · 2024-12-02
> >
> > Dear Reviewer UP1g,
> >
> > We sincerely hope that our response has addressed your concerns. If you have any further questions or concerns, please do not hesitate to let us know. We greatly appreciate and look forward to your further feedback.
> >
> > Best regards,
> >
> > The Authors

---

> > ### Comment · Reviewer_UP1g · 2024-12-02
> >
> > ***CallBack***
> >
> > Thank you to the author for addressing some of my questions.
> >
> > - Why does HiDiffusion have higher hardware requirements? It avoids using global attention at the highest-resolution latent and instead uses window attention, which theoretically consumes less memory than ScaleCrafter and AccDiffusion. I hope to see HiDiffusion's performance at even higher resolutions.
> >
> > - These training-free high-resolution generation methods are much slower than SDXL+BSRGAN, yet they fail to show significant advantages in objective metrics. What is the practical value of such methods? I noticed the visual comparisons with SDXL+BSRGAN in the supplementary materials, but these may be cherry-picked and not representative of the overall performance. Based on objective metrics, SDXL+BSRGAN demonstrates highly competitive results compared to these training-free methods (Table 8) and is unquestionably the superior choice in terms of computational cost (Table 9).

---

> ### Author Response · Authors · 2024-11-29
> **Gentle Reminder**
>
> Dear Reviewer UP1g,
>
> Thank you once again for your comprehensive and thoughtful feedback on our submission. As the discussion period nears its end, we are eager to know if our additional results and clarifications have adequately addressed your questions.
> We would sincerely appreciate any further perspectives or discussions you might have at this stage.
> Thank you for your time and engagement!
>
> Best regards,
>
> Authors

---

> ### Author Response · Authors · 2024-12-03
> **Response to Reviewer UP1g**
>
> We sincerely appreciate your time and constructive feedback. Below, we would like to address your concerns systematically and comprehensively.
>
> # Response
>
> 1. We would like to note that we encountered an OOM issue when conducting HiDiffusion experiments to generate 4096X4096 resolution images by strictly following their official implementation. We speculate that this may be related to some memory optimization tricks. We have tried some memory-saving tricks and are able to run it now. We will update the corresponding results soon. Hopefully, we can get the results before rebuttal period closes. Regardless of whether we can meet the deadline, we will certainly include the relevant results in the next version of the paper.
>
> 2. We would like to note that quantitatively evaluating the generative models is challenging. Though we can use metrics such as FID and IS, it is widely acknowledged that these metrics fail to comprehensively evaluate the performance of model’s generation. As a result, user studies are commonly employed to provide human-level evaluation with more intuition[1-7]. For example, in ScaleCrafter, they conducted both quantitative and user study analyses in comparison with the SD+SR approach. Their results show that, although ScaleCrafter performs worse than SD+SR on quantitative metrics, users significantly prefer the textures and details generated by ScaleCrafter. One important reason is that the goal of the SR model is to produce images consistent with the input, which limits its performance in high-resolution generation – needing more detail for true high-resolution visuals beyond simple smoothing[6-11]. For the rigor of the paper, following your suggestion, we are currently conducting a user study to compare SDXL+BSRGAN and AP-LDM. Due to time constraints, we will present the results in the next version of the paper.
>
> # References
>
> [1] Are GANs Created Equal? A Large-Scale Study. NeurIPS 2018.\
> [2] High-Resolution Image Synthesis with Latent Diffusion Models. CVPR 2022.\
> [3] SDXL: Improving Latent Diffusion Models for High-Resolution Image Synthesis. ICLR 2024.\
> [4] Adding Conditional Control to Text-to-Image Diffusion Models. ICCV 2023.\
> [5] AnimateDiff: Animate Your Personalized Text-to-Image Diffusion Models without Specific Tuning. ICLR 2024.\
> [6] ScaleCrafter: Tuning-free Higher-Resolution Visual Generation with Diffusion Models. ICLR 2024.\
> [7] Training-free Diffusion Model Adaptation for Variable-Sized Text-to-Image Synthesis. NeurIPS 2023.\
> [8] DemoFusion: Democratising High-Resolution Image Generation With No $$$. CVPR 2024.\
> [9] AccDiffusion: An Accurate Method for Higher-Resolution Image Generation. ECCV 2024.\
> [10] Make a Cheap Scaling: A Self-Cascade Diffusion Model for Higher-Resolution Adaptation. ECCV 2024.\
> [11] Any-Size-Diffusion: Toward Efficient Text-Driven Synthesis for Any-Size HD Images. AAAI 2024.

---

> ### Author Response · Authors · 2024-12-03
> **Model Comparison at 4096x4096 Resolution**
>
> We maintained the same settings as in the experiments described in the main text and tested the performance of HiDiffusion in generating **$4096\times 4096$** images, as shown in the table below (the best results are marked in **bold**, and the second-best results are marked in *italics*):
>
> | Methods      | FID      | IS       | FID$_c$  | IS$_c$   | CLIP     |
> | ------------ | -------- | -------- | -------- | -------- | -------- |
> | SDXL         | 191.4    | 8.3      | 114.1    | 12.4     | 22.9     |
> | MultiDiff.   | 168.4    | 6.5      | 76.6     | *14.4*   | 23.1     |
> | ScaleCrafter | 164.5    | 9.4      | 170.1    | 7.3      | 22.3     |
> | UG           | 187.3    | 7.0      | 197.6    | 6.3      | 21.8     |
> | DemoFusion   | 101.4    | 20.7     | 63.5     | 13.5     | *24.7*   |
> | AccDiff.     | 103.2    | 20.1     | 65.9     | 13.3     | 24.6     |
> | SDXL+BSRGAN  | **90.0** | *20.9*   | **56.0** | 13.8     | **25.2** |
> | HiDiff.      | 144.1    | 12.5     | 147.0    | 7.4      | 21.2     |
> | AP-LDM       | *90.6*   | **21.1** | *59.0*   | **14.8** | 24.6     |
>
> From the table, we can observe that HiDiffusion's performance drops  when synthesizing higher-resolution images. We speculate that this is due to the following reasons: \
> (1) HiDiffusion directly performs upsampling and downsampling on the UNet feature maps, leading to significant distribution shifts;  \
> (2) HiDiffusion uses windowing, which forces a change in the sequence length of the attention, resulting in a severe shift in the entropy of the attention map.
>
> We hope our supplementary experiments address your concerns. If you have any further questions, please do not hesitate to let us know before the rebuttal period ends. Your feedbacks are highly appreciated and helpful.
>
> (We are currently conducting experiments at 2048x4096 and 4096x2048 resolutions. Hopefully, we can update these results before rebuttal period closes.)

---

> ### Author Response · Authors · 2024-12-04
> **Comparison of Generation Results at 2048x4096 and 4096x2048 Resolutions**
>
> We maintained the same settings as in the experiments described in the main text and tested the performance of HiDiffusion in generating **$2048\times 4096$** and **$4096\times 2048$** images.
>
> - The comparison of generation results at **$2048\times 4096$** resolution is shown in the table below (the best results are marked in **bold**, and the second-best results are marked in *italics*):
>
> | Methods      | FID      | IS       | FID$_c$  | IS$_c$   | CLIP     |
> | ------------ | -------- | -------- | -------- | -------- | -------- |
> | SDXL         | 149.9    | 9.5      | 106.3    | 12.0     | 24.4     |
> | MultiDiff.   | 125.8    | 9.6      | 71.9     | *15.7*   | 24.6     |
> | ScaleCrafter | 161.9    | 10.0     | 154.3    | 7.5      | 23.3     |
> | UG           | 155.7    | 8.2      | 165.0    | 6.6      | 21.7     |
> | DemoFusion   | 96.3     | 17.7     | 62.3     | 15.0     | *25.0*   |
> | AccDiff.     | 95.5     | 16.4     | 62.9     | 11.1     | 24.5     |
> | SDXL+BSR.    | **80.7** | *19.8*   | **50.2** | 12.3     | **25.1** |
> | HiDiff.      | 120.7    | 12.2     | 93.0     | 13.6     | 24.2     |
> | AP-LDM       | *89.0*   | **20.3** | *56.0*   | **19.0** | *25.0*   |
>
> - The comparison of generation results at **$4096\times 2048$** resolution is shown in the table below (the best results are marked in **bold**, and the second-best results are marked in *italics*):
>
> | Methods      | FID      | IS       | FID$_c$  | IS$_c$   | CLIP     |
> | ------------ | -------- | -------- | -------- | -------- | -------- |
> | SDXL         | 173.1    | 9.1      | 108.5    | 11.5     | 23.9     |
> | MultiDiff.   | 149.0    | 9.0      | 70.5     | 14.4     | *24.4*   |
> | ScaleCrafter | 175.1    | 9.7      | 167.3    | 8.0      | 21.6     |
> | UG           | 185.3    | 6.8      | 175.7    | 6.2      | 20.5     |
> | DemoFusion   | 99.6     | 16.4     | 61.9     | *14.7*   | *24.4*   |
> | AccDiff.     | 102.2    | 15.2     | 65.4     | 11.5     | 24.2     |
> | SDXL+BSR.    | **92.7** | *17.6*   | *57.9*   | 12.1     | **24.9** |
> | HiDiff.      | 128.4    | 12.8     | 98.3     | 11.3     | 23.1     |
> | AP-LDM       | *93.2*   | **19.5** | **56.9** | **16.5** | **24.9** |
>
> - From the results in the two tables above, we can see that HiDiffusion's performance drops at higher resolutions, which is consistent with the trend observed in the **$4096\times 4096$** resolution results we reported earlier. We speculate that the reasons for the performance drop in these higher resolutions are the same:
>     - HiDiffusion directly performs upsampling and downsampling on the UNet feature maps, leading to significant distribution shifts;
>     - HiDiffusion uses windowing, which forces a change in the sequence length of the attention, resulting in a severe shift in the entropy of the attention map.
>
> We hope that the complete results we have provided could address your concerns. We greatly appreciate the time and effort you have dedicated to reviewing our work. Your suggestions have been extremely helpful in improving our work.

---

### Official Review · Reviewer_38vj · 2024-11-03

**Soundness:** 3
**Presentation:** 3
**Contribution:** 3
**Rating:** 6
**Confidence:** 4

**Summary:**

In this article, a new method called AP-LDM is introduced. This method is both user-friendly and efficient, and it can generate higher-quality high-resolution (HR) images while speeding up the generation process without the need for prior training. AP-LDM divides the noise removal process during image generation into two steps:
The first step is dedicated to training noise reduction for resolution enhancement, referred to as "meticulously trained resolution denoising." The purpose of this step is to utilize a new method called Attenive Guidance, which employs a new parameter-free self-attention technique, to make the generated image structures more consistent.
The second step is "progressive enhancement to high-resolution denoising." In this step, the image's pixel resolution is increased iteratively to eliminate potential image errors or blurriness that may arise from the resolution enhancement process.

**Strengths:**

1. The author demonstrates the effectiveness of the proposed method through comparative and ablation experiments.
2. Through experimental means, the author discovers issues with adoption in the latent space, and therefore proposes a novel upsampling method in pixel space that can avoid significant loss of texture details.

**Weaknesses:**

1. There is no explanation for why attention mechanisms can improve structural consistency in noisy spaces.

**Questions:**

1. Have you considered adjusting the guidance scale in a learnable way?
2. In line 207, the author points out that the early noise in denoising is non semantic, so is the noise semantic in other processes? Is it meaningful to perform attention calculation on noise?
3. The author mentioned in the contribution that it provides a speed of up to 5 times in HR image generation, but did not address why there is a speed improvement compared to other models. It is hoped that further explanation can be provided.

**Details Of Ethics Concerns:**

I have no concerns.

---

> ### Author Response · Authors · 2024-11-28
> **Response to Reviewer 38vj**
>
> # About Supplementary Materials
> Our full paper file exceeds 100MB. Due to upload size limitations, we have submitted a compressed version of the paper. To ensure clear visualization of the images, **we have uploaded pages 18–27 of the appendix as supplementary materials**. Please refer to these pages for further details!
>
> # **Response to weaknesses**
>
> In the revised paper, Appendix A.5 provides a detailed explanation of how PFSA improves structural consistency. Specifically, **PFSA can cluster related tokens in the latent representation**, which improves the structural consistency of the latent representation. At the same time, during the early and mid-stages of denoising, **PFSA amplifies the high-frequency components of the latent representation**, which results in generated images with richer details and colors. Please refer more details in Appendix A.5.
>
> # **Response to Questions**
>
> - **Response to Q1**: Thank you for your constructive feedback. In fact, we have considered using gradients to optimize the guidance scale. The idea was to **freeze the parameters of SDXL and learn the guidance scale**. Specifically, we can sample a time step t from {1, …, T} and sample an image x_0, then optimize the guidance scale by predicting the noise added to x_0. It is worth noting that attentive guidance improves the image **by adjusting the latent representation across multiple consecutive time steps**. Intuitively, the guidance scale should be **optimized across different time steps simultaneously**, but achieving this is quite challenging. Currently, we have not found a suitable framework for learning the guidance scale.
>
> - **Response to Q2**: A lot of research has shown that diffusion models generate images by first producing the low-frequency components and then the high-frequency components[1, 2]. As the denoising process progresses, the low-frequency structural information in the latent representation gradually increases, and the semantics become more pronounced. When t becomes large, due to the noise present and the fact that the representation is in the latent space, the semantic information is difficult to interpret directly. In the revised paper, in Appendix A.5, we provide a detailed explanation of PFSA. Through experiments, we demonstrate that PFSA can cluster tokens in the latent representation, even when noise is present (Figure 15). Additionally, we performed a Fourier transform on the latent representations at different time steps (Figure 16). From the Fourier transform, it is evident that when PFSA is applied in the early denoising process, the resulting latent representation exhibits a significant frequency band difference from the original latent representation. Later in the denoising process, applying PFSA causes only minor changes in the high-frequency components. This indirectly suggests that the early latent representations differ semantically from the later ones.
>
> - **Response to Q3**: For each denoising step, **the time consumption on high-resolution image generation stage is several times longer than that on low-resolution image** generation stage. Methods like DemoFusion require the entire denoising process to be applied to high-resolution image generation stage, making them extremely time-consuming. Although there are differences in detail between high and low-resolution images, the low-frequency structures are consistent. AP-LDM fully leverages this property, **allowing the denoising process for high-resolution images to avoid starting from scratch and instead only require 5-15 steps**, thus significantly accelerating the generation process. In the revised version of our paper, in Appendix A.4, we provide detailed information on the time consumption for denoising at different resolutions.
>
> # Reference
> [1] FreeDoM: Training-Free Energy-Guided Conditional Diffusion Model. ICCV 2023. \
> [2] Relay Diffusion: Unifying diffusion process across resolutions for image synthesis. ICLR 2024.

---

> > ### Author Response · Authors · 2024-12-02
> >
> > Dear Reviewer 38vj,
> >
> > We sincerely hope that our response has addressed your concerns. If you have any further questions or concerns, please do not hesitate to let us know. We greatly appreciate and look forward to your further feedback.
> >
> > Best regards,
> >
> > The Authors

---

> ### Author Response · Authors · 2024-11-29
> **Gentle Reminder**
>
> Dear Reviewer 38vj,
>
> Thank you once again for your comprehensive and thoughtful feedback on our submission. As the discussion period nears its end, we are eager to know if our additional results and clarifications have adequately addressed your questions.
> We would sincerely appreciate any further perspectives or discussions you might have at this stage.
> Thank you for your time and engagement!
>
> Best regards,
>
> Authors

---

### Official Review · Reviewer_XiPQ · 2024-11-03

**Soundness:** 2
**Presentation:** 2
**Contribution:** 2
**Rating:** 6
**Confidence:** 4

**Summary:**

This paper introduces a framework aimed at enhancing high-resolution image generation without the need for additional training. The AP-LDM model decomposes the denoising process into two stages: attentive training-resolution denoising and progressive high-resolution denoising. The first stage employs a parameter-free self-attention mechanism to improve structural consistency in latent representations, while the second stage progressively upsamples images in pixel space, mitigating artifacts typically introduced by latent space upsampling. Experimental results demonstrate that AP-LDM achieves up to a 5× speedup in generation time while maintaining high image quality. This approach not only accelerates the generation process but also enhances the overall efficiency of high-resolution image synthesis.

**Strengths:**

+ The paper introduces AP-LDM, a training-free method that enhances high-resolution image generation while speeding up the process and provides quantitative and qualitative comparison results with other works.
+ A simple attentive guidance module is proposed, which is developed upon a parameter-free self-attention mechanism that improves structural consistency in latent representations.
+ A progressive upsampling strategy is employed in pixel space, which reduces artifacts associated with latent space upsampling

**Weaknesses:**

-	The proposed designs are informed by empirical observations and necessitate attentive guidance and progressive upsampling for optimal hyperparameter selection in weighted feature fusion and temporal interference.
-	PFSA functions as self-modulated attention, enhancing features with strong responses. However, potential side effects of this design remain unaddressed in the limitations section.
-	In Section 4.4 and Figure 13, the method is referred to as "SwiftFusion." Consistency in terminology should be maintained throughout the paper.

**Questions:**

The paper employs progressive upsampling with either one or two sub-stages. It would be beneficial to present intermediate results from each sub-stage, as pixel-level decoded images are available at each step. Additionally, will content consistency be maintained as the number of sub-stages increases, for example, to three or more stages?

---

> ### Author Response · Authors · 2024-11-29
> **Reply to Reviewer XiPQ**
>
> # About Supplementary Materials
> Our full paper file exceeds 100MB. Due to upload size limitations, we have submitted a compressed version of the paper. To ensure clear visualization of the images, **we have uploaded pages 18–27 of the appendix as supplementary materials**. Please refer to these pages for further details!
>
> # **Response to Weaknesses**
>
> 1. Although hyperparameter selection is empirical as in the most previous methods, we conduct comprehensive ablation experiments to provide a specific range of parameter selection, demonstrating our generalization capacity across different prompts and frameworks.
>     - We followed most of training-free high-resolution generation methods with wide recognition, such as DemoFusion, AccDiffusion, and HiDiffusion, to tune hyperparameter experimentally. We would like to highlight that **the key lies not in whether parameters are tuned empirically, but in whether the parameters derived from ablation studies are applicable to most scenarios**.
> In our experiments, we identified a suitable range of parameters for attentive guidance and progressive upsampling when using the SDXL model. Through extensive evaluations, we demonstrated that **this hyperparameter range is effective across various resolutions**.
> In the revised version of our paper, Appendix A.8 further shows that **the hyperparameter range, suitable for SDXL, also applies to SD2.1**. In Appendix A.7, we demonstrate that **the hyperparameters for attentive guidance also perform well for AP-LDM and other generative frameworks (DemoFusion and HiDiffusion)**. These experiments **highlight the generalizability of the hyperparameter selection derived through experimental exploration**.
>     - Since there are variations in individual preferences for image contrast and detail richness, we note that **there is no absolute standard for selecting the optimal parameters** of the attentive guidance and the progressive upsampling. **We have provide a stable range of parameter choices and explained how each parameter influences the generation results**, where the specific selection of parameters ultimately depends on individual preferences.
>
> 2. **Existing training-free generative frameworks confront two main challenges**: (1) difficulty in generating images with rich details and vibrant colors, and (2) excessively long generation times. To address these issues, we introduce two methods: (1) PFSA, which enhances the structural consistency of latent representations, improving the richness of details and colors in generated images; and (2) a progressive pixel-space upsampling framework, enabling the rapid generation of high-resolution images with globally consistent semantics. \
> We emphasize that **the issue of text generation errors, mentioned in the Limitation section, is not caused by our PFSA but rather stems from the inherent limitations of the pretrained model's generative capabilities**. As shown in Figure 11, **even when directly using SDXL for inference at its trained resolution, the generated images still exhibit such text errors**.
> Since PFSA operates only during the low-resolution generation process, **it introduces negligibly additional inference time**. Even though our AP-LDM takes longer to generate ultra-high-resolution images, it is **still faster than existing methods**.
>
> 3. Regarding the inconsistency in naming in the paper, we have corrected the relevant typographical errors and performed a thorough review. We sincerely appreciate your feedback.
>
> # **Response to the Questions:**
>
> Considering that **our primary objective is fundamentally a HR image generation task rather than traditional super-resolution tasks**, it is sufficient to maintain the consistency of the target subject **without requiring perfect alignment of the details**.\
> We present the intermediate results of the model's generation process. In Figure 14 of the revised paper, we showcase the intermediate outputs. The results in Figure 14 were obtained using a total of three sub-stages. As observed, **the images across different stages exhibit high structural consistency, with only minor and reasonable differences in details**.

---

> > ### Author Response · Authors · 2024-12-02
> >
> > Dear Reviewer XiPQ,
> >
> > We sincerely hope that our response has addressed your concerns. If you have any further questions or concerns, please do not hesitate to let us know. We greatly appreciate and look forward to your further feedback.
> >
> > Best regards,
> >
> > The Authors

---

> > ### Comment · Reviewer_XiPQ · 2024-12-02
> >
> > Thank you to the authors for their responses and the additional experimental results. Some of my concerns have been addressed through their clarifications. I would like to revise my score to a 6.

---

> > > ### Author Response · Authors · 2024-12-02
> > >
> > > Dear reviewer XiPQ,
> > >
> > > Thank you for your thoughtful response and positive feedback. We truly appreciate the time and effort you’ve dedicated to reviewing our work and are glad that our clarifications and additional experiments have addressed your concerns. Your insights are valuable in improving our work.
> > >
> > > Sincerely,
> > >
> > > Authors

---

> ### Author Response · Authors · 2024-11-29
> **Gentle Reminder**
>
> Dear Reviewer XiPQ,
>
> Thank you once again for your comprehensive and thoughtful feedback on our submission. As the discussion period nears its end, we are eager to know if our additional results and clarifications have adequately addressed your questions.
> We would sincerely appreciate any further perspectives or discussions you might have at this stage.
> Thank you for your time and engagement!
>
> Best regards,
>
> Authors

---

### Author Response · Authors · 2024-11-28
**Global Response to Reviews**

We sincerely thank all the reviewers for their insightful feedback and for recognizing the strengths of our work. Specifically, we appreciate their acknowledgment of the innovation of our simple yet efficient attentive guidance (Reviewers XiPQ, UP1g, and 9bp6), the extensive experiment analysis (Reviewers UP1g, 38vj, and 9bp6), the significant advantages in both generation quality and efficiency (Reviewers XiPQ, UP1g, and 9bp6), and the impressive texture details of generated HR images (Reviewer XiPQ and 38vj). We will comprehensively address the specific concerns of each reviewer in detail within our subsequent responses, including more details on the motivation, inner mechanism of the attentive guidance, extra experiments, and some discussions.

---

> ### Author Response · Authors · 2024-11-28
> **About Supplementary Material**
>
> Our full paper file exceeds 100MB. Due to upload size limitations, we have submitted a compressed version of the paper.
>
> To ensure clear visualization of the images, we have uploaded pages 18–27 of the appendix as supplementary materials. Please refer to these pages for further details!

---

### Meta-Review · Area_Chair_H9VP · 2024-12-20

**Metareview:**

The paper presents a new approach for high-resolution image generation and does not require training.  The proposed method divides the denoising process into two stages: attentive training-resolution denoising and progressive high-resolution denoising. While the method shows some improvement in performance, its theoretical contribution in terms of innovation is limited, making it slightly below the acceptance threshold.

**Additional Comments On Reviewer Discussion:**

Reviewers have expressed concerns regarding the innovation of the method, the completeness of the experiments, and the performance of the results. The authors' rebuttal addressed most of these concerns, providing additional clarification and experimental results. However, despite these efforts, the reviewers ultimately assigned scores of three slightly above the acceptance threshold and one slightly below the threshold.

---

### Decision · Program_Chairs · 2025-01-22

Reject